# SAFER: Risk-Constrained Sample-then-Filter in Large Language Models

**Qingni Wang**[1], **Yue Fan**[1], **Xin Eric Wang**[1,2*]

[1]University of California, Santa Cruz
[2]University of California, Santa Barbara
{qwang158, yfan71}@ucsc.edu, ericxwang@ucsb.edu

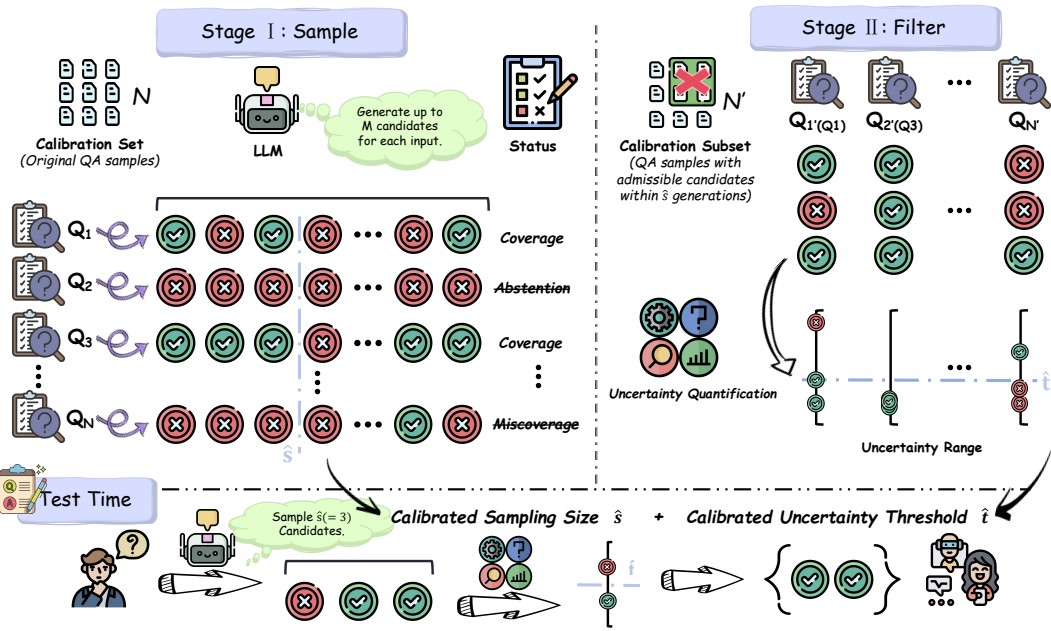

Figure 1: Overview of SAFER's calibration and test-time process. In Stage I, we derive a statistically valid minimum sample budget $\hat{s}$ that can strictly control the test-time risk of the candidate set of size $\hat{s}$ not covering correct answers. In Stage II, we employ the calibration instances, which can obtain admissible answers within $\hat{s}$ samples, to calibrate a threshold $\hat{t}$. This threshold filters out unreliable answers in the candidate set while still constraining the miscoverage risk of the final prediction set.

## Abstract

As large language models (LLMs) are increasingly deployed in risk-sensitive applications such as real-world open-ended question answering (QA), ensuring the trustworthiness of their outputs has become critical. Existing selective conformal prediction (SCP) methods provide statistical guarantees by constructing prediction sets with a constrained miscoverage rate for correct answers. However, prior works unrealistically assume that admissible answers for all instances can be obtained via finite sampling, even for open-ended QA scenarios that lack a fixed and finite solution space. To address this, we introduce a two-stage risk control framework comprising abstention-aware **SA**mpling and conformalized **F**ilt**ER**ing (SAFER). Firstly, on a held-out calibration set, SAFER calibrates a sampling budget within the maximum sampling cap, using the Clopper–Pearson exact method at a user-desired risk level (i.e., the maximum allowable miscoverage rate of the sampling sets). If the risk level cannot be satisfied within the cap, we abstain; otherwise, the calibrated sampling budget becomes the minimum requirements at

---

*Corresponding author

test time. Then, we employ calibration instances where correct answers are attainable under the calibrated budget and apply the conformal risk control method to determine a statistically valid uncertainty threshold, which filters unreliable distractors from the candidate set for each test data point. In this stage, SAFER introduces an additional risk level to guide the calculation of the threshold, thereby controlling the risk of correct answers being excluded. We evaluate SAFER on three free-form QA datasets utilizing five popular LLMs, and demonstrate that it rigorously constrains two-stage miscoverage risks at test time. Furthermore, we show that SAFER is compatible with various task-specific admission criteria and calibration-test split ratios, highlighting its robustness and high data efficiency. Our project page is available at this link.

# 1 INTRODUCTION

Recent advances in large language models (LLMs) have propelled substantial progress across various question answering (QA) tasks (Alawwad et al., 2025; Yang et al., 2025). Nevertheless, critical challenges persist—most notably, the prevalence of hallucinated content and miscalibration of confidence scores associated with model outputs (Huang et al., 2025; Cox et al., 2025). These issues compromise the reliability of LLMs and constrain their deployment in real-world conversational systems (Guo et al., 2025; Singhal et al., 2025; Liu et al., 2025a; Li et al., 2025; 2024). Uncertainty quantification (UQ) has emerged as a promising solution to these concerns, by providing quantitative indicators of predictive uncertainty (Duan et al., 2024; Wang et al., 2024c; 2025c). It can trigger user-facing warnings if the uncertainty exceeds a predefined threshold, thereby enhancing the trustworthiness of query results.

Split conformal prediction (SCP) is a distribution-free, model-agnostic framework that provides rigorous UQ (Papadopoulos et al., 2002; Angelopoulos & Bates, 2021; Angelopoulos et al., 2024a). It guarantees the coverage of ground-truth labels in classification tasks, assuming data exchangeability. Specifically, SCP defines a nonconformity measure for each data point as the residual between the model prediction and the ground-truth on a held-out calibration set, and then calibrates an uncertainty threshold under a user-specified risk level (rigorous uncertainty score), which guides test-time construction of prediction sets by selecting high-quality classes. The probability of each data-driven set failing to cover the true class is controlled by the risk level (upper bound of risk). While recent work has effectively applied SCP to multiple-choice question answering (MCQA) (Ye et al., 2024), its application to open-ended settings remains limited due to two fundamental challenges: ❶ Open-ended QA lacks a fixed output space, and we cannot ensure that for each given inquiry, the deployed LLM is capable of generating admissible answers via finite sampling (Wang et al., 2024c; 2025a; Kaur et al., 2024). ❷ Due to the temperature settings of sampling or hallucinations, the candidate (or sampling) set may contain irrelevant and unreliable answers (Wang et al., 2025a;d), which prevents them from being presented to users for downstream decision-making (Cresswell et al., 2024; 2025).

To resolve the dual challenges and control the miscoverage risk in open-ended QA tasks, we propose a two-stage calibration framework consisting of abstention-aware **SA**mpling and conformalized **F**ilt**ER**ing (SAFER), as shown in Figure 1. First, SAFER adopts a learn-then-test (LTT) paradigm to derive the required sampling budget at test time on the calibration set under a user-specified risk level $\alpha$ (Angelopoulos et al., 2025), while developing an abstention mechanism (Yadkori et al., 2024). Specifically, instead of assuming that every sampling set contains correct answers (Wang et al., 2024c; Kaur et al., 2024; Wang et al., 2025a), SAFER imposes an upper bound $M$ on the number of samples per question, and deems that if no admissible answers are obtained within $M$ attempts, the model abstains from it. Then, we apply the Clopper–Pearson exact method (Clopper & Pearson, 1934; Wang et al., 2025b) to establish an upper confidence bound of the miscoverage rate at each sampling budget, and determine the minimum number of samples necessary for the test-time risk—the probability of failing to include admissible answers—to remain below $\alpha$. If the calibration procedure shows that this risk bound cannot be satisfied even with $M$ samples, SAFER abstains, ensuring that the framework never produces invalid prediction sets under unreachable risk levels.

Once the abstention-aware sampling stage secures the desired coverage (or abstains when unattainable), the resulting candidate set undergoes conformalized filtering, which further prunes unreliable predictions and provides more efficient references for users. Specifically, SAFER first evaluates the

uncertainty score of each sampled answer in the candidate set and removes those with scores above a certain threshold. Since heuristic uncertainty measures cannot perfectly distinguish between correct and incorrect answers (Wang et al., 2025c), we leverage the conformal risk control (CRC) framework (Angelopoulos et al., 2024b) and introduce another risk level $\beta$ to guide the calibration of the threshold, thereby controlling the risk of incorrectly excluding correct answers while removing unreliable answers from each candidate set. This two-stage design ensures that SAFER delivers reliable prediction sets while remaining robust to the inherent unpredictability of open-ended QA.

We evaluate SAFER on three open-ended QA benchmarks across five open-source LLMs. Experimental results demonstrate that SAFER validly estimates the upper bound of the miscoverage rate at each sampling budget and constrains the test-time risks under various user-specified risk levels ($\alpha$). Through the conformalization of uncertainty guided by another risk level ($\beta$), SAFER validly filters out unreliable candidates and maintains the statistical rigor of prediction sets. Moreover, we validate the robustness of SAFER under four correctness evaluation criteria, as well as its efficiency in achieving risk control on the test set using only a small amount of calibration data. Our contributions can be summarized as follows:

- We introduce SAFER as a novel risk control framework that provides user-specified coverage of admissible answers in open-ended QA tasks.

- We incorporate the abstention mechanism into the sampling stage and develop a more practical method for calibrating the sampling budget.

- We conformalize the uncertainty of admissible answers on the calibration set, and calibrate a threshold to filter out unreliable candidates within each sampling set, which maintains the statistical validity of the prediction sets and enhances predictive efficiency.

## 2 RELATED WORK

**Uncertainty quantification.** Informing users of the model's uncertainty score within the QA process can enhance the reliability of downstream decision-making (Liu et al., 2025b; Xia et al., 2025; Shorinwa et al., 2025). Previous studies have developed probabilistic frameworks (Hou et al., 2025; Wang et al., 2025c) like semantic entropy (SE) Kuhn et al. (2023); Farquhar et al. (2024) or prompted models to express their uncertainty (Rathi et al., 2025; Xu et al., 2025). However, these approaches are heuristic and lack statistical verification. SCP can transform any heuristic notion of uncertainty from any model into a statistically rigorous one in closed-ended tasks (Kumar et al., 2023; Ye et al., 2024; Kostumov et al., 2024). Yet, due to the inherent definition of nonconformity in SCP, detailed in Appendix B, applying it to open-ended tasks remains an open topic.

**SCP in open-ended QA.** In white-box scenarios, SE-SCP (Kaur et al., 2024; Kuhn et al., 2023) establishes the nonconformity based on the entropy derived from semantically clustered predictions. CLM (Quach et al., 2024) utilizes the LTT framework (Angelopoulos et al., 2025) to identify parameter configurations with rigorous statistical guarantees. Under black-box conditions, LofreeCP (Su et al., 2024; Gupta et al., 2022) and ConU (Wang et al., 2024c) constructs nonconformity measures by employing Self-Consistency-Based (SCB) uncertainty estimates, such as sampling frequency. TRON (Wang et al., 2025a) further employs sampling size calibration to determine the required number of samples at test time and then defines nonconformity scores based on frequency, thereby achieving two-stage risk control. However, existing research either assumes that every instance can yield correct answers via finite sampling or fails to consider that certain risk levels may be infeasible.

**Learn-then-test.** LTT (Angelopoulos & Bates, 2021; Angelopoulos et al., 2025; 2024a) is a modular post-hoc calibration framework that reframes risk control as a multiple hypothesis testing problem: it first learns a predictive model, then systematically searches over a low-dimensional parameter space using calibration data, testing each candidate through valid concentration-based p-values and applying family-wise error rate control to identify parameters that satisfy user-specified risk levels (Ni et al., 2025; Jung et al., 2025), thereby yielding finite-sample statistical guarantees independent of model architecture or data distribution. In this work, we draw on the principles of LTT to analyze, on the calibration set, whether to abstain under a given risk level or to search for the minimal sampling size that can statistically guarantee the test-time coverage of admissible answers.

## 3  METHODOLOGY

### 3.1  OVERVIEW

Following SCP-based frameworks, we hold out a set of $N$ calibration samples $\mathcal{D}_{cal} = \{(x_i, y_i^*)\}_{i=1}^N$, where $x_i \in \mathcal{X}$ denotes the input question of the $i$-th calibration data and $y_i^* \in \mathcal{Y}$ is the corresponding ground-truth answer. Let $F : \mathcal{X} \to \mathcal{Y}$ denote an LLM. For each question, we sample multiple (e.g., $s$) candidate answers $\{\hat{y}_j^{(i)}\}_{j=1}^s$. Let $A : \mathcal{Y} \times \mathcal{Y} \to [0, 1]$ denote a task-specific relevance function that evaluates how well the generated answer matchs the ground-truth for a given input. An answer $\hat{y}_j^{(i)} \in \mathcal{Y}$ is considered admissible only if $A(\hat{y}_j^{(i)}, y_i^*) \geq \lambda_A$, where $\lambda_A$ is a user-specified relevance threshold. For each sample, up to $M$ candidate answers are allowed to be drawn. If none of the $M$ sampled answers is admissible (i.e., $\forall \hat{y} \in \{\hat{y}_j^{(i)}\}_{j=1}^M, A(\hat{y}, y_i^*) < \lambda_A$), we treat it as an abstention, attributing this to the limitations of the LLM's capabilities. Moreover, to filter out unreliable candidates from each sampling set, we denote by $U : \mathcal{Y} \to [0, 1]$ an uncertainty measure that estimates the sentence entropy of the sampled answer (Quach et al., 2024; Duan et al., 2024). An answer is retained only if its uncertainty score falls below a threshold $t$.

Our goal is to calibrate two parameters on the calibration set, $s$ and $t$: the former guides the sampling process, while the latter filters out irrelevant answers from the sampled set, enabling the construction of a prediction set $\mathcal{C}$ for each test input $x_{test} \in \mathcal{X}$ that covering at least one admissible answer with high probability. Formally, given a desired error tolerance $\epsilon$, we provide the following guarantee:

$$\Pr\Big(\Pr\big(\forall \hat{y} \in \mathcal{C}(x_{test}), A(\hat{y}, y_{test}^*) < \lambda_A\big) \leq \epsilon\Big) \geq 1 - \delta, \tag{1}$$

where $\delta$ is the significance level (e.g., 0.05), and the outer probability controls for the sensitivity of our algorithm with respect to calibration data.

### 3.2  ABSTENTION-AWARE SAMPLING BUDGET CALIBRATION

We begin by calibrating a statistically valid sampling budget that guarantees the coverage of admissible answers. For a given sampling budget $s$, we first report the number of calibration data points for which the candidate sets of size $s$ fail to include any admissible answers:

$$\hat{m}_{cal}(s) = \sum_{i=1}^N \mathbf{1}\left\{\forall \hat{y} \in \left\{\hat{y}_j^{(i)}\right\}_{j=1}^s, A(\hat{y}, y_i^*) < \lambda_A\right\}, \tag{2}$$

and then estimate the empirical miscoverage rate on the calibration set:

$$\hat{r}_{cal}(s) = \frac{\hat{m}_{cal}(s)}{N}. \tag{3}$$

We aim to establish an upper bound of the true (or system) miscoverage rate on new test samples at sampling budget $s$, denoted as $R(s) := \Pr\big(\forall \hat{y} \in \mathcal{C}(x), A(\hat{y}, y^*) < \lambda_A\big)$, through realizations observed on the calibration set. To achieve this goal, we leverage the Clopper-Pearson exact method, by constructing a high-probability (i.e., at least $1 - \delta$) exact upper confidence bound for $R(s)$:

$$\hat{R}^+(s) = \sup\left\{R : \Pr\big(\mathrm{Bin}(N, R) \leq \lceil N\hat{r}_{cal}(s)\rceil\big) \geq \delta\right\}, \tag{4}$$

which characterizes the largest system risk at a sampling budget of $s$ that does not contradict the empirical error rate $\hat{r}_{cal}(s)$ observed from the calibration set. That is, if the system risk $R(s)$ exceeds the upper bound $\hat{R}^+(s)$, then observing an empirical error rate as low as $\hat{r}_{cal}(s)$ constitutes a low-probability event, and is considered statistically implausible at significance level $\delta$. Hence, we obtain the follow guarantee:

$$\Pr(R(s) \leq \hat{R}^+(s)) \geq 1 - \delta. \tag{5}$$

A formal proof of Eq. 5 is provide in Appendix B.

To rigorously ensure the high-probability coverage of admissible answers via sampling $s$ candidates, we calibrate $s$ by constraining the upper bound $\hat{R}^+(s)$ to fall below a user-specified risk level $\alpha$:

$$\hat{s} = \inf\left\{s \in [1, M] : \hat{R}^+(s) \leq \alpha\right\}, \tag{6}$$

By construction, this choice of $\hat{s}$ minimizes the sampling cost subject to the risk control requirement. Note that if $\hat{R}^+(M) > \alpha$, we determine that, given the limited sampling, it is not possible to control the miscoverage rate under the current risk level $\alpha$, and thus abstain from making a prediction. For a valid sampling budget $\hat{s}$, given that $\hat{R}^+(\hat{s}) \geq R(\hat{s})$ holds with high probability, we have

$$\Pr(R(\hat{s}) \leq \hat{R}^+(\hat{s}) \leq \alpha) \geq 1 - \delta, \quad \hat{s} \in [1, M]. \tag{7}$$

We then obtain the test-time guarantee in the sampling stage:

$$\Pr\Big(\forall \hat{y} \in \{\hat{y}_j^{(test)}\}_{j=1}^{\hat{s}}, A\big(\hat{y}, y_{test}^*\big) < \lambda_A\Big) \leq \alpha, \quad \hat{s} \in [1, M], \tag{8}$$

with probability at least $1 - \delta$ over the draw of the calibration set $\mathcal{D}_{\mathrm{cal}}$.

### 3.3 CONFORMALIZED FILTERING

After determining the sampling budget $\hat{s}$, since each model's sampling space inevitably contains other unreliable answers, directly delivering the calibrated candidate set to users may not be suitable for downstream decision-making due to these distractors. To address this issue while maintaining statistical rigor, SAFER introduces another risk level $\beta$ and leverages the conformal risk control framework Angelopoulos et al. (2024b): it conformalizes the uncertainty scores of reliable answers on the calibration set, thereby calibrating a statistically rigorous uncertainty threshold that is employed to filters out unreliable candidates from each test sample's candidate set at test time to form the final prediction set. To maintain risk constraints, we employ the conformal risk control framework Angelopoulos et al. (2024b) to calibrate a statistically rigorous threshold to identify high-quality answers. Specifically, we first select, from the calibration set $\mathcal{D}_{cal}$, all samples that contain at least one admissible answer within their $\hat{s}$-sized sampling set to form the calibration subset:

$$\mathcal{D}_{cal}(\hat{s}) = \Big\{(x_i, y_i^*) \in \mathcal{D}_{cal} : \exists \hat{y} \in \{\hat{y}_j^{(i)}\}_{j=1}^{\hat{s}}, A(\hat{y}, y_i^*) \geq \lambda_A\Big\}. \tag{9}$$

Here, we aim to ensure that under the same sampling budget $\hat{s}$ as set for test samples, each calibration data point includes at least one admissible answer in its candidate set. Then, this calibration subset of size $N' = |\mathcal{D}_{cal}(\hat{s})|$ can be utilized to calibrate the threshold that differentiates admissible answers from unreliable ones, subject to the other target risk level $\beta$.

For a given threshold $t$, we construct a prediction set for each calibration sample in $\mathcal{D}_{cal}(\hat{s})$:

$$\mathcal{C}_t(x_i) = \Big\{\hat{y} \in \{\hat{y}_j^{(i)}\}_{j=1}^{\hat{s}} : U(\hat{y}) \leq t\Big\}, \quad (x_i, y_i^*) \in \mathcal{D}_{cal}(\hat{s}). \tag{10}$$

$U(\hat{y})$ calculatethe accumulation of the token-wise entropy over the whole sentence $\hat{y}$:

$$U(\hat{y}) = \sum_k -\log p(z_k \mid \hat{y}_{<k}), \tag{11}$$

where $p(z_k \mid \hat{y}_{<k})$ denotes the probability of generating $z_k$ as the $k$-th token and $\hat{y}_{<k}$ refers to the previously generated tokens within $\hat{y}$.

We then define a loss function as:

$$l_i(t) = \mathbf{1}\Big\{\forall \hat{y} \in \mathcal{C}_t(x_i), A(\hat{y}, y_i^*) < \lambda_A\Big\}. \tag{12}$$

Let $L_{N'}(t) = \frac{1}{N'} \sum_{i=1}^{N'} l_i(t)$ denote the average loss over all samples in the calibration subset. We determine the uncertainty threshold as

$$\hat{t} = \inf\Big\{t : \frac{N' L_{N'}(t) + 1}{N' + 1} \leq \beta\Big\} = \inf\Big\{t : L_{N'}(t) \leq \frac{\beta(N' + 1) - 1}{N'}\Big\}. \tag{13}$$

By applying $\hat{t}$ in test-time filtering, we can ensure that if the $\hat{s}$-sized candidate set of each test sample contains at least one admissible answer, the probability of the calibrated prediction set not covering admissible answers remains below $\beta$, formulated as:

$$\Pr\Big(\forall \hat{y} \in \mathcal{C}_{\hat{t}}(x_{test}), A(\hat{y}, y_{test}^*) < \lambda_A\Big) \leq \beta, \quad \text{if } \exists \hat{y} \in \{\hat{y}_j^{(test)}\}_{j=1}^{\hat{s}}, A(\hat{y}, y_{test}^*) \geq \lambda_A. \tag{14}$$

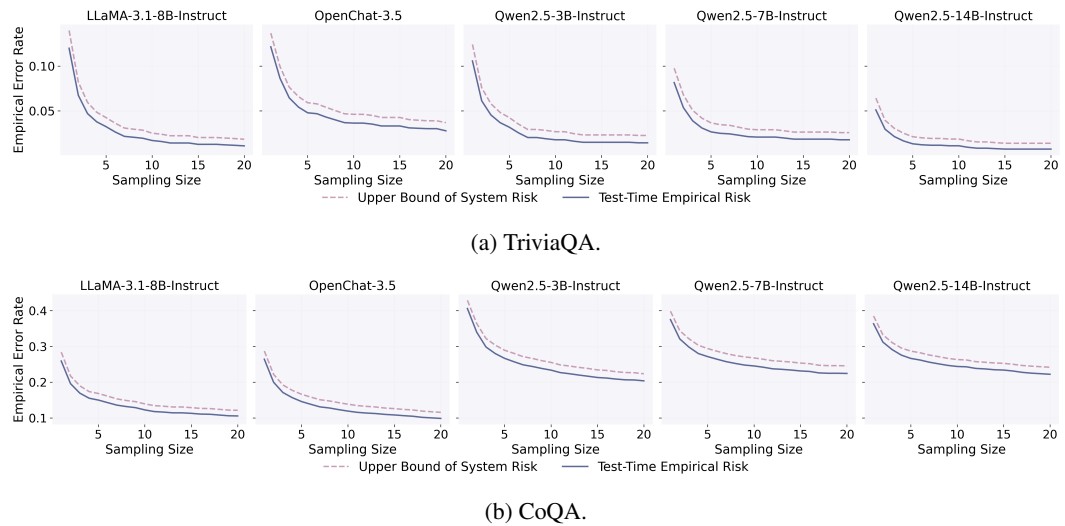

(a) TriviaQA.

(b) CoQA.

Figure 2: Empirical miscoverage rates under different sampling budgets. The dashed lines denote the system miscoverage upper bounds derived via the Clopper–Pearson exact method on the calibration set, while the solid lines show the corresponding empirical miscoverage rates on the test set.

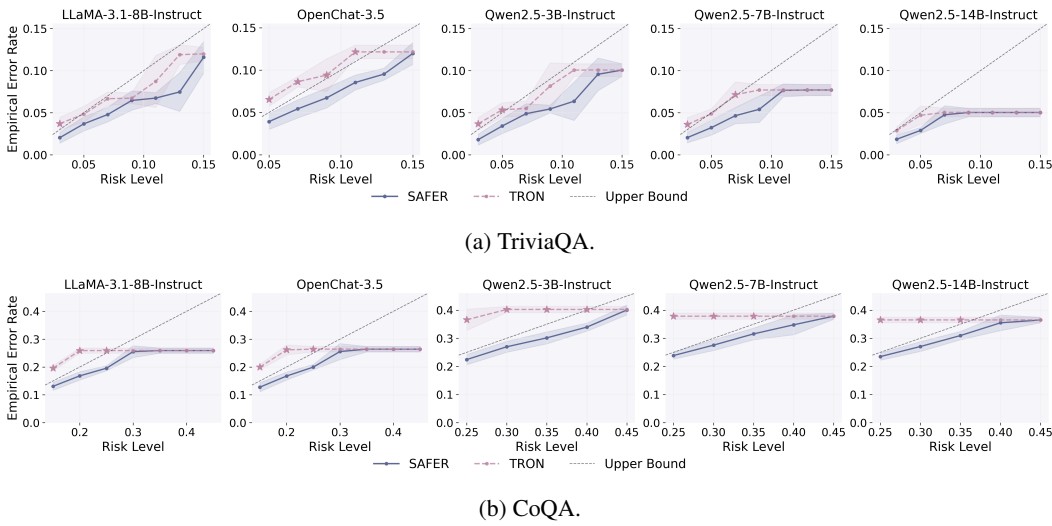

(a) TriviaQA.

(b) CoQA.

Figure 3: Comparison of TRON and SAFER on the control of test-time EER in the sampling stage.

Of course, in the absence of an admissible answer in the candidate set, the calibrated prediction set will inevitably lack one as well.

Finally, we establish formal guarantees on the coverage of admissible answers:

$$\Pr\Big(\Pr\big(\forall \hat{y} \in \mathcal{C}_{\hat{t}}(x_{test}), A(\hat{y}, y_{test}^*) < \lambda_A\big) \leq \alpha + \beta - \alpha\beta\Big) \geq 1 - \delta. \tag{15}$$

The complete proofs of Eq. 14 and Eq. 15 are provided in Appendix B.

## 4 EXPERIMENT

### 4.1 EXPERIMENT SETTINGS

**Benchmarks.** We evaluate SAFER on three open-ended QA benchmarks, categorized into closed-book settings: TriviaQA (Joshi et al., 2017) and ScienceQA (Lu et al., 2022), and open-book settings

Table 1: Test-time EER results in the filtering stage on the TriviaQA ($\alpha = 0.05$) and CoQA ($\alpha = 0.25$) dataset at various risk levels ($\beta$).

(a) TriviaQA ($\alpha = 0.05$)

| $\beta$ | 0.05 | 0.10 | 0.15 | 0.20 | 0.25 | 0.30 | 0.35 | 0.40 |
|---|---|---|---|---|---|---|---|---|
| Upper Bound ($\alpha + \beta - \alpha\beta$) | 0.0975 | 0.1450 | 0.1925 | 0.2400 | 0.2875 | 0.3350 | 0.3825 | 0.4300 |
| LLaMA-3.1-8B-Instruct | 0.0844±0.0137 | 0.1322±0.0160 | 0.1805±0.0189 | 0.2265±0.0198 | 0.2753±0.0215 | 0.3225±0.0211 | 0.3698±0.0198 | 0.4175±0.0195 |
| OpenChat-3.5 | 0.0890±0.0128 | 0.1375±0.0155 | 0.1851±0.0175 | 0.2318±0.0185 | 0.2800±0.0211 | 0.3274±0.0232 | 0.3752±0.0221 | 0.4225±0.0214 |
| Qwen2.5-3B-Instruct | 0.0822±0.0145 | 0.1295±0.0189 | 0.1757±0.0193 | 0.2223±0.0204 | 0.2709±0.0212 | 0.3186±0.0231 | 0.3683±0.0241 | 0.4125±0.0236 |
| Qwen2.5-7B-Instruct | 0.0802±0.0136 | 0.1264±0.0158 | 0.1738±0.0172 | 0.2207±0.0194 | 0.2693±0.0209 | 0.3184±0.0211 | 0.3663±0.0229 | 0.4140±0.0241 |
| Qwen2.5-14B-Instruct | 0.0771±0.0104 | 0.1253±0.0146 | 0.1746±0.0165 | 0.2223±0.0184 | 0.2692±0.0212 | 0.3161±0.0217 | 0.3612±0.0217 | 0.4100±0.0221 |

(b) CoQA ($\alpha = 0.25$)

| $\beta$ | 0.05 | 0.10 | 0.15 | 0.20 | 0.25 | 0.30 | 0.35 | 0.40 |
|---|---|---|---|---|---|---|---|---|
| Upper Bound ($\alpha + \beta - \alpha\beta$) | 0.2875 | 0.3250 | 0.3625 | 0.4000 | 0.4375 | 0.4750 | 0.5125 | 0.5500 |
| LLaMA-3.1-8B-Instruct | 0.2347±0.0122 | 0.2743±0.0155 | 0.3139±0.0171 | 0.3546±0.0168 | 0.3954±0.0196 | 0.4347±0.0212 | 0.4729±0.0212 | 0.5142±0.0204 |
| OpenChat-3.5 | 0.2373±0.0119 | 0.2794±0.0153 | 0.3201±0.0189 | 0.3598±0.0196 | 0.4010±0.0205 | 0.4401±0.0224 | 0.4802±0.0222 | 0.5203±0.0228 |
| Qwen2.5-3B-Instruct | 0.2625±0.0179 | 0.3001±0.0210 | 0.3389±0.0242 | 0.3778±0.0253 | 0.4159±0.0250 | 0.4549±0.0278 | 0.4932±0.0278 | 0.5324±0.0277 |
| Qwen2.5-7B-Instruct | 0.2761±0.0149 | 0.3145±0.0181 | 0.3521±0.0186 | 0.3912±0.0176 | 0.4276±0.0175 | 0.4640±0.0195 | 0.5015±0.0207 | 0.5383±0.0207 |
| Qwen2.5-14B-Instruct | 0.2710±0.0137 | 0.3076±0.0157 | 0.3469±0.0168 | 0.3851±0.0188 | 0.4245±0.0186 | 0.4629±0.0201 | 0.5023±0.0213 | 0.5395±0.0205 |

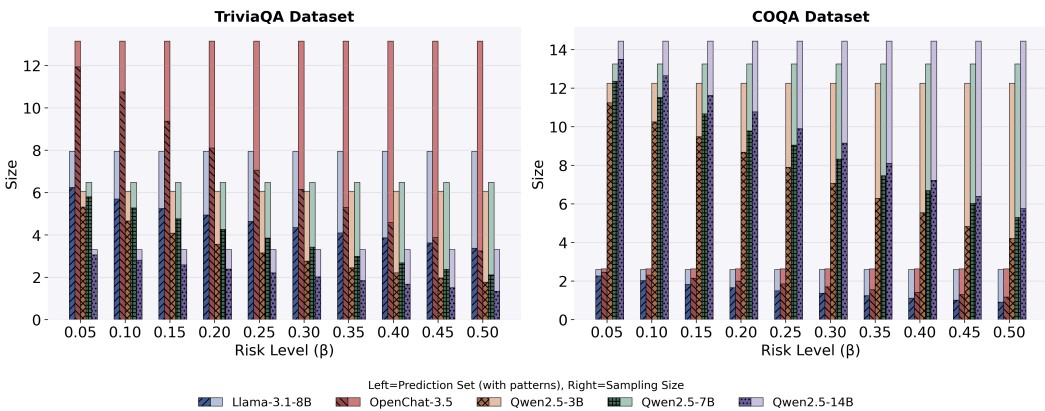

Figure 4: Comparison of the (average) calibrated sampling budget in the sampling stage with the (average) prediction set size after filtering out unreliable answers in the second stage. Filtering compresses the calibrated budgets into **tighter sets** while maintaining risk control.

with passage context: CoQA (Reddy et al., 2019). Additionally, we evaluate generalization on the multi-modal reasoning benchmark MMVet Yu et al. (2023) in Appendix E.5. More details can be found in Appendix D.1.

**Backbone LLMs.** We consider three popular series of "off-the-shelf" LLMs—OpenChat (Wang et al., 2024a): OpenChat-3.5, LLaMA (AI@Meta, 2024): LLaMA-3.1-8B-Instruct, and Qwen (Yang et al., 2024): Qwen-2.5-3B-Instruct, Qwen-2.5-7B-Instruct, and Qwen-2.5-14B-Instruct.

**Correctness Evaluation.** We deem a candidate admissible if the task-specific relevance function $R$ indicates agreement with the reference beyond a predefined threshold. We utilize sentence similarity (Wang et al., 2024c; 2025d) with a threshold of 0.6 by default. Moreover, we consider Rouge-L score (Lin, 2004; Duan et al., 2024), bi-entailment based on natural language inference (NLI) (Wang et al., 2025c; Kuhn et al., 2023), and LLM-based semantic evaluation (Zhang et al., 2024). Details of implementation are provided in Appendix D.2.

**Evaluation Metric.** We evaluate the statistical rigor of SAFER by checking whether the test-time Empirical Error Rate (EER) (Angelopoulos & Bates, 2021; Wang et al., 2024c; 2025a;d) is bounded by the user-specified risk level. In the sampling stage, EER represents the average miscoverage rate

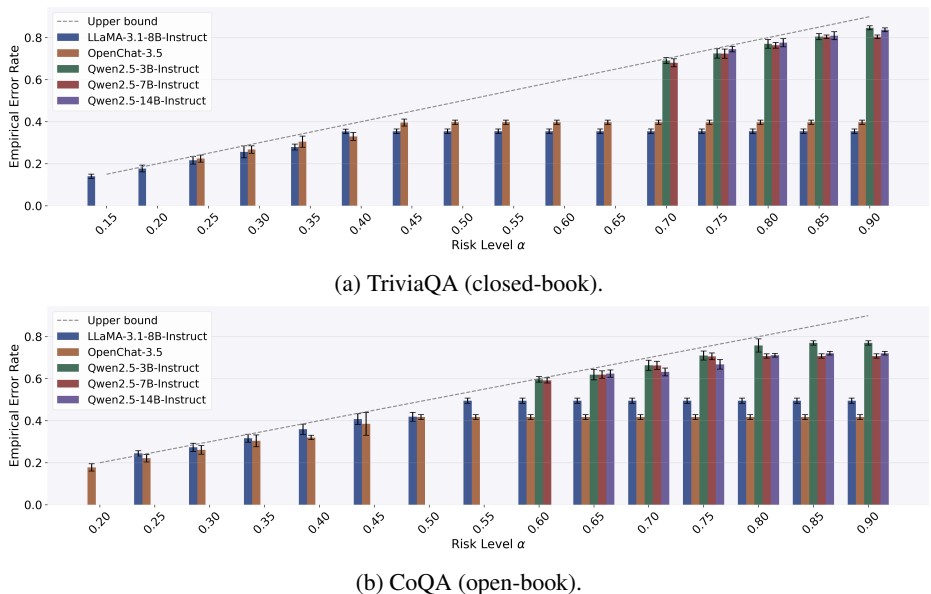

(a) TriviaQA (closed-book).

(b) CoQA (open-book).

Figure 5: Test-time EER results in the sampling stage with Rouge-L score as the correctness metric.

over all the candidate sets of size $\hat{s}$ on the test set (size $N_t$), denoted as

$$\text{EER} = \frac{1}{N_t} \sum_{i_t=1}^{N_t} \mathbf{1} \left\{ \forall \hat{y} \in \{\hat{y}_j^{(i_t)}\}_{j=1}^{\hat{s}}, A\big(\hat{y}, y_{i_t}^*\big) < t_R \right\}.$$

In the filtering stage, EER is computed on the prediction sets formed by answers selected from the sampling set according to the calibrated uncertainty threshold, denoted as

$$\text{EER} = \frac{1}{N_t} \sum_{i_t=1}^{N_t} \mathbf{1} \left\{ \forall \hat{y} \in \mathcal{C}_{\hat{t}}(x_{i_t}), A(\hat{y}, y_{i_t}^*) < t_R \right\}.$$

**Hyperparameters.** Following (Wang et al., 2025d;b), we employ multinominal sampling for candidate answers with the generating temperature of 1.0. The generation length of free-form answers is fixed at 36 for TriviaQA and CoQA, and 24 for ScienceQA. We set the significance level $\delta$ to 0.05. Moreover, we set the split ratio of the calibration and test amples to 0.5 by default. We provide a detailed analysis of the impact of the sampling budget $M$ in Appendix E.4.

## 4.2 EMPIRICAL EVALUATIONS

We first verify the statistical validity of the system risk upper bound established in Eq. 4 utilizing the Clopper–Pearson exact method. As demonstrated in Figure 2, across both the TriviaQA and CoQA datasets, the empirical miscoverage rates of five LLMs on the test set consistently remain below the corresponding confidence bounds under different sampling budgets. The results indicate that the statistically rigorous sampling budget derived from the calibration set—together with the upper bound of its corresponding EER—effectively carries over to test time. Then, we can pre-compute a high-confidence upper bound on the test-set EER for a given sampling budget using statistical methods on the calibration data, which establishes the foundation for calibrating the sampling budget.

We next discuss how, under various desired risk levels ($\alpha$), SAFER controls the EER of the sampling stage. As presented in Figure 3, we compare our method with the previous TRON approach under conditions with abstention, i.e., we maintain the test data points that fail to obtain admissible answers via finite sampling ($M$ samples in this case). It can be observed that, at certain risk levels, the sampling budget derived by TRON fails to control the EER on the test set. For instance, when $\alpha = 0.03$, TRON with the OpenChat-3.5 model on TriviaQA yields a test-time EER exceeding 0.06. In contrast, SAFER consistently maintains the EER below the corresponding risk level $\alpha$ across

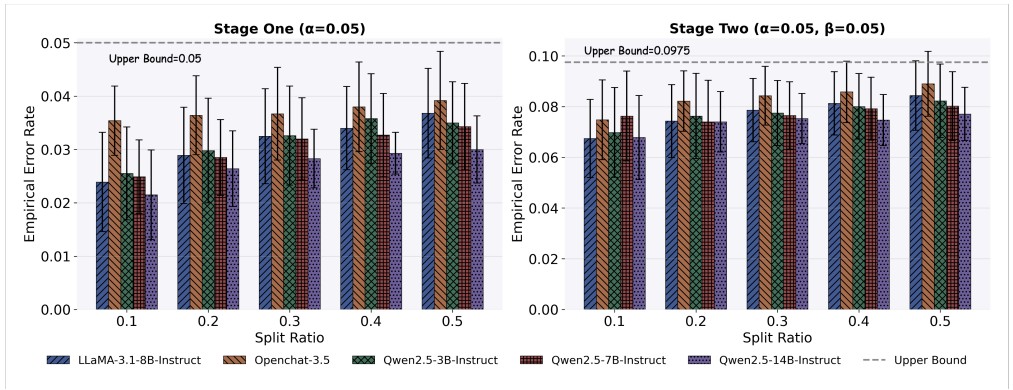

Figure 6: Test-time EER results in both the sampling and filtering stages across various calibration-test split ratios on the TriviaQA dataset with sentence similarity as the correctness metric.

both datasets and all five LLMs. Since TRON does not account for cases where the model fails to obtain an admissible answer through limited sampling during calibration, the derived sampling budget cannot be effectively applied to the test set. As a result, some test data points that should be abstained from lead to errors exceeding the target risk level. SAFER effectively resolves this issue, making the sampling stage more rigorous and practical.

As mentioned, for the same question, LLMs may produce some irrelevant or incorrect answers in the candidate set due to hallucinations or the use of temperature sampling strategies (Zhu et al., 2025; Hou et al., 2025). SAFER adapts the conformal risk control framework to address this. As illustrated in Table 1, although some correct answers may be filtered out due to the limitations of UQ (as no perfect method yet exists to fully distinguish correct from incorrect answers), SAFER still maintains control over the miscoverage rate of admissible answers in the final prediction sets on both the TriviaQA and CoQA datasets—ensuring that the EER consistently remains below the joint upper bound of $\alpha$, and $\beta$. At that point, SAFER achieves two-stage guarantees, rigorously controlling test-time EER. Furthermore, as presented in Figure 4, given a fixed sampling budget under a specified $\alpha$, the filtering guided by the risk level $\beta$ substantially reduces the size of the final prediction set. For instance, when $\beta$ is as small as 0.1, the average set size decreases from 7.9 to 5.5 on the TriviaQA dataset with the Llama-3.1-8B model, and the prediction set size continues to shrink as $\beta$ increases. At the same time, the test-time EER consistently remains below the upper bound. Results of two-stage risk control on ScienceQA are provided in Appendix E.2. While we primarily utilize entropy-based uncertainty, our framework is metric-agnostic. We validate SAFER's effectiveness on black-box models (e.g., GPT-4o-mini) using consistency-based frequency scores in Appendix E.3.

## 4.3 SENSITIVITY ANALYSES

**Sensitivity to Correctness Metric.** SAFER applies to various correctness evaluation criteria. In this section, we discuss the case of using the Rouge-L score. As shown in Figure 5, SAFER still manages to keep the test-time EER within the target risk level on TriviaQA and CoQA. It is worth noting that since Rouge-L score evaluates the longest common subsequence of sentences, in the open-book setting, where the context provides the exact lexical form of the reference answer, relatively weaker LLMs can achieve lower risk levels on CoQA. By contrast, in the closed-book setting, semantically equivalent but lexically different answers may occur, making this alignment less straightforward. Results of risk control in the filtering stage on the TriviaQA and CoQA datasets using Rouge-L score, along with results under other evaluation metrics, are provided in Appendix E.1.

**Sensitivity to Calibration-Test Split Ratio.** SAFER calibrates the sampling budget and uncertainty threshold on the calibration set. As shown in Figure 6, when the split ratio between calibration and test data decreases from 0.5 to 0.1, the EER on the test set consistently remains below the user-specified upper bound. This demonstrates that SAFER can achieve risk control over a large number of test samples with only a small amount of calibration data, highlighting its efficiency.

## 5 CONCLUSION

In this paper, we propose a two-stage risk control framework, SAFER, which provides user-specified coverage of admissible answers in open-domain QA tasks. SAFER adopts the LTT paradigm, allowing that some instances may fail to yield a correct answer within finite sampling. It first determines the minimal sampling size that meets the user-specified risk level. If the risk level cannot be satisfied even under the maximal sampling budget, SAFER abstains, which effectively ensures the statistical rigor of the prediction sets delivered for downstream decision-making, while overcoming the drawbacks of existing methods that rely on strong assumptions. SAFER further adapts the conformal risk control framework to uncertainty-threshold calibration, which, while maintaining risk control, effectively removes unreliable candidates from the sampled sets and produces prediction sets with higher predictive efficiency. We envision SAFER as a general-purpose framework to advance trustworthy, uncertainty-aware decision-making in foundation models across diverse downstream tasks.

### LIMITATION

Although SAFER addresses the control of miscoverage under conditions where a correct answer cannot be obtained through limited sampling, bypassing the assumption that all problems for a deployed model can be correctly answered with finite sampling, and achieves two-stage miscoverage risk control, it still relies on the assumption of data exchangeability. In the case of distribution shifts, SAFER would need further optimization to ensure risk control on the test set.

### ETHICS STATEMENT

This work introduces SAFER, a risk-constrained framework for uncertainty quantification in large language models. Our study does not involve human or animal subjects, personal or sensitive data, or any interventions that may raise ethical concerns. All datasets employed (TriviaQA, CoQA, and ScienceQA) are publicly available and widely used in the research community. We strictly adhere to their original licenses and usage policies. While our framework aims to improve the reliability of open-ended question answering, we acknowledge the potential misuse of large language models in sensitive domains. By providing statistically rigorous guarantees, SAFER is designed to enhance trustworthiness and mitigate harmful mispredictions, thus promoting responsible deployment of AI systems.

### REPRODUCIBILITY STATEMENT

We have taken multiple steps to ensure reproducibility of our results.

**Datasets.** We provide detailed descriptions of the datasets and experimental splits in Appendix D.1.

**Algorithms and Proofs.** The methodology is described with precise mathematical notation in Section 3, and all theoretical guarantees are proved in Appendix C.

**Hyperparameters.** Evaluation hyperparameters, including sampling budgets, thresholds, and generation settings, are reported in Section 4.1.

**Code and Implementation.** An anonymous code repository with implementation and instructions will be made available after the review process, once the paper is accepted and published.

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

## A  USE OF LARGE LANGUAGE MODELS

We used LLMs only as general-purpose writing assistants, specifically to check grammar, spelling, and style consistency. The LLMs did not contribute to research ideation, experimental design, data analysis, or result interpretation. All substantive content and scientific contributions were developed solely by the authors.

## B  BACKGROUND OF SPLIT CONFORMAL PREDICTION IN CLASSIFICATION TASKS

We summarize Split Conformal Prediction (SCP) for multi-class classification with precise notation. Let $\mathcal{D}_{\text{cal}} = \{(x_i, y_i^*)\}_{i=1}^n$ be an i.i.d. *calibration set* of size $n$, where $y_i^* \in [K] = \{1, \ldots, K\}$ is the ground-truth class, and let $\hat{f} : \mathcal{X} \to \Delta^{K-1}$ be a fixed *pretrained classifier* that maps an input $x$ to class probabilities $\hat{f}(x) = (\hat{f}(x)_1, \ldots, \hat{f}(x)_K)$ with $\sum_{k=1}^K \hat{f}(x)_k = 1$. Define the *nonconformity score* for each calibration example as $s_i = 1 - \hat{f}(x_i)_{y_i^*}$ (one minus the predicted probability of the true class), and sort the scores in ascending order, $s_{(1)} \leq \cdots \leq s_{(n)}$. Given a target error rate $\alpha \in (0, 1)$, set the finite-sample $(1 - \alpha)$ quantile

$$\hat{q} = s_{(\lceil (n+1)(1-\alpha) \rceil)}.$$

At test time, for a new input $x$, form the conformal prediction set

$$C_\alpha(x) = \{ y \in [K] : 1 - \hat{f}(x)_y \leq \hat{q} \}.$$

Under exchangeability between $\mathcal{D}_{\text{cal}}$ and the test point $(X, Y^*)$, SCP attains the marginal coverage guarantee

$$\mathbb{P}\{ Y^* \in C_\alpha(X) \} = \frac{\lceil (n+1)(1-\alpha) \rceil}{n+1} \geq 1 - \alpha.$$

## C  PROOFS

In this section, we first provide a complete proof that the upper confidence bound $\hat{R}_{1-\delta}^+(s)$ defined in Eq. 4 satisfies $\Pr\left(R(s) \leq \hat{R}_{1-\delta}^+(s)\right) \geq 1 - \delta$ defined in Eq. 5.

Without loss of generality, we define $\hat{R}(s)$ as the empirical error rate variable on the calibration set with the sampling size of $s$. We then define the CDF of $\hat{R}(s)$ under the true risk $R(s)$ as :

$$D(r \mid R(s)) = \Pr(\hat{R}(s) \leq r \mid R(s)) \tag{16}$$

This CDF characterizes the likelihood that the empirical error rate is no greater than a specified risk threshold $r$, under the true distribution governed by $R(s)$.

We then define the corresponding inverse CDF as

$$D^{-1}(p) = \sup \{r : D(r \mid R(s)) \leq p\} \tag{17}$$

By the definition of $\hat{R}^+(s)$, it holds that

$$D\left(\hat{r}_{cal}(s) \mid \hat{R}^+(s)\right) = \delta. \tag{18}$$

Since $D^{-1}(\delta) = \sup \{r : D(r \mid R(s)) \leq \delta\}$, if $D(\hat{r}_{cal}(s) \mid R(s)) < \delta$, then we have $\hat{r}_{cal}(s) < D^{-1}(\delta)$. This implies the following equivalence

$$\left\{ D(\hat{r}_{cal}(s) \mid R(s)) < \delta \Rightarrow \hat{r}_{cal}(s) < D^{-1}(\delta) \right\}$$
$$\Longleftrightarrow \left\{ \Pr(D(\hat{r}_{cal}(s) \mid R(s)) < \delta) \leq \Pr(\hat{r}_{cal}(s) < D^{-1}(\delta)) \right\}$$

Considering that $D(r \mid R(s))$ is monotonic decreasing in $R(s)$, i.e., larger $R(s)$ leads to smaller $D(r \mid R(s))$, if $R(s) > \hat{R}^+(s)$, then $D(\hat{r}_{cal}(s) \mid R(s)) < \delta$. This implies

$$\left\{ R(s) > \hat{R}^+(s) \Rightarrow D(\hat{r}_{cal}(s) \mid R(s)) < \delta \right\}$$
$$\Longleftrightarrow \left\{ \Pr\left(R(s) > \hat{R}^+(s)\right) \leq \Pr(D(\hat{r}_{cal}(s) \mid R(s)) < \delta) \right\}$$

Thus,

$$
\begin{aligned}
\Pr\left(R\left(s\right) \le \hat{R}^{+}\left(s\right)\right) &= 1 - \Pr\left(R\left(s\right) > \hat{R}^{+}\left(s\right)\right) \\
&\ge 1 - \Pr\left(D\left(\hat{r}_{cal}\left(s\right) \mid R\left(s\right)\right) < \delta\right), \\
&\ge 1 - \Pr\left(\hat{r}_{cal}\left(s\right) < D^{-1}\left(\delta\right)\right) \\
&\ge 1 - \delta
\end{aligned}
\tag{19}
$$

demonstrating that $\hat{R}^{+}\left(s\right)$ is the upper endpoint of the Clopper–Pearson exact $1 - \delta$ confidence interval for the $R\left(s\right)$. This completes the proof.

Next, we prove the theoretical soundness of Eq. 14 and Eq. 15.

By the exchangeability of $N'$ calibration data points and the given test instances, we have
$$
l_{test}(t) \backsim \mathrm{Uniform}\left(\{l_1(t), \cdots, l_{N'}(t), l_{test}(t)\}\right).
$$

Given that $l_{test}(\hat{t}) \le 1\ (\in \{0, 1\})$, we obtain
$$
\begin{aligned}
\mathbb{E}\left(l_{test}(\hat{t})\right) &= L_{N'+1}\left(\hat{t}\right) \\
&= \frac{1}{N'+1} \sum_{i=1}^{N'+1} l_i(\hat{t}) \\
&= \frac{N' L_{N'}(\hat{t}) + l_{test}(\hat{t})}{N'+1} \\
&\le \frac{N' L_{N'}(\hat{t}) + 1}{N'+1} \\
&\le \alpha.
\end{aligned}
\tag{20}
$$

Then, if the sampling set of size $\hat{s}$ contains admissible answers, i.e.,
$$
\exists \hat{y} \in \{\hat{y}_j^{(test)}\}_{j=1}^{\hat{s}}, A(\hat{y}, y_{test}^*) \ge \lambda_A,
$$
we guarantee the error rate of the calibrated prediction set of the test data point failing to cover admissible answers
$$
\begin{aligned}
\Pr\left(\forall \hat{y} \in \mathcal{C}_{\hat{t}}\left(x_{test}\right), A(\hat{y}, y_{test}^*) < \lambda_A\right) &= \mathbb{E}\left(\mathbf{1}\left\{\forall \hat{y} \in \mathcal{C}_t\left(x_i\right), A(\hat{y}, y_i^*) < \lambda_A\right\}\right) \\
&= \mathbb{E}\left(l_{test}(\hat{t})\right) \\
&\le \beta
\end{aligned}
\tag{21}
$$

Since $\Pr\left(\forall \hat{y} \in \{\hat{y}_j^{(test)}\}_{j=1}^{\hat{s}}, A\left(\hat{y}, y_{test}^*\right) < \lambda_A\right) \le \alpha$ is satisfied with probability at least $1 - \delta$, we achieve two-stage risk control and guarantee the upper bound of the miscoverage rate:
$$
\begin{aligned}
&\Pr\left(\forall \hat{y} \in \mathcal{C}_{\hat{t}}(x_{test}), A(\hat{y}, y_{test}^*) < \lambda_A\right) \\
&= \Pr\left(\forall \hat{y} \in \mathcal{C}_{\hat{t}}\left(x_{test}\right), A(\hat{y}, y_{test}^*) < \lambda_A \mid \exists \hat{y} \in \{\hat{y}_j^{(test)}\}_{j=1}^{\hat{s}}, A(\hat{y}, y_{test}^*) \ge \lambda_A\right) + \\
&\quad \Pr\left(\forall \hat{y} \in \{\hat{y}_j^{(test)}\}_{j=1}^{\hat{s}}, A(\hat{y}, y_{test}^*) < \lambda_A\right) \\
&\le 1 - (1 - \alpha)(1 - \beta) \\
&\le \alpha + \beta - \alpha\beta
\end{aligned}
\tag{22}
$$
with probability at least $1 - \delta$ over the draw of the calibration set $\mathcal{D}_{\mathrm{cal}}$. This completes the proof.

## D  DETAILS OF EXPERIMENTAL SETTINGS

### D.1  DETAILS OF DATASETS

**TriviaQA**    (Joshi et al., 2017) is a large-scale open-domain QA dataset with about 650K question–answer pairs, paired with automatically retrieved evidence from Wikipedia and the Web. We adopt the closed-book setting, evaluating models only on questions and answers. For our experiments, we randomly sample 2000 data points from the validation set. Promot example is shown in Figure 14.

**CoQA** (Reddy et al., 2019) is a conversational QA dataset with more than 127K question–answer pairs from dialogues across seven domains. Answers are free-form spans grounded in source passages, and later questions depend strongly on dialogue history. We use CoQA in the open-book setting, where models have access to both the context. We evaluate on 2,000 instances sampled from the validation set. Prompt example is shown in Figure 15.

**ScienceQA** (Lu et al., 2022) is a science-related open-domain QA dataset designed to evaluate the ability of models to understand complex scientific text. It contains approximately 13,679 crowd-sourced science questions. In our experiments, we use the full validation set consisting of 1,000 questions. Prompt example is shown in Figure 16.

**MMVet** (Yu et al., 2023) is a comprehensive benchmark designed to evaluate the integrated capabilities of large multimodal models. It consists of 217 diverse image-text samples that require models to solve complex tasks by combining core vision-language skills. We utilize MMVet to assess the generalization of our framework to reasoning-intensive multi-modal scenarios. Prompt example is shown in Figure 17.

### D.2 DETAILS OF CORRECTNESS METRICS

Rouge-L deems a sampled answer as admissible if its longest common subsequence, regarding the ground truth answer, is larger than a threshold. Considering that there are many semantically equivalent but differently phrased answers in open-ended QA tasks, we set the threshold of Rouge-L as 0.3 by default (Duan et al., 2024). We also adopt a bidirectional entailment approach to evaluate the correctness of free-form answers Kuhn et al. (2023); Farquhar et al. (2024); Lin et al. (2024); Wang et al. (2025c). Specifically, we employ an off-the-shelf DeBERTa-large model He et al. (2021) as the Natural Language Inference (NLI) classifier, which outputs logits over three semantic relation classes: entailment, neutral, and contradiction. An answer is deemed correct if the classifier predicts entailment for both directions, i.e., when evaluated on (answer $\rightarrow$ ground-truth) and (ground-truth $\rightarrow$ answer). Moreover, following Zhang et al. (2024), we employ Qwen-2.5-3B-Instruct to assess whether the answer matches the ground-truth, using the prompt template:

> Ground truth: <sentence 1>.
> Model answer: <sentence 2>.
> Please verify if the model answer matches the ground truth. Respond with either 'Correct' or 'Wrong' only.

## E ADDITIONAL EXPERIMENTAL RESULTS

### E.1 GENERALIZATION UNDER ALTERNATIVE EVALUATION CRITERIA

**ROUGE-L Evaluation** In the section 4.2, we have already examined Rouge-L as an alternative correctness criterion in the sampling stage on TriviaQA and CoQA, showing that the calibrated sampling procedure maintains bound-valid risk control. Here, we extend the analysis to the filtering stage across TriviaQA and CoQA. As shown in Tables 2b and 2a, EER under Rouge-L evaluation remain strictly below the theoretical bound $\alpha + \beta - \alpha\beta$ across models and risk levels, with EER increasing smoothly as $\beta$ grows. These results confirm that SAFER consistently achieves valid risk control under Rouge-L, demonstrating its robustness to task-specific evaluation metrics.

**Entailment Evaluation** We next examine entailment-based correctness, where a candidate is deemed admissible if it is bidirectionally entailed with the reference answer at a threshold of 0.5. Figure 7 reports the sampling stage results across TriviaQA and CoQA. We observe that empirical error rates remain consistently below the upper bound $\alpha$ across all risk levels. These results confirm that the abstention-aware sampling procedure generalizes robustly under entailment correctness. Table 3 summarizes the filtering stage performance under the entailment criterion. Across all datasets and models, EER values remain strictly below the combined upper bound $\alpha + \beta - \alpha\beta$ at each setting. This shows that SAFER maintains reliable two-stage guarantees under an entailment-based definition of correctness.

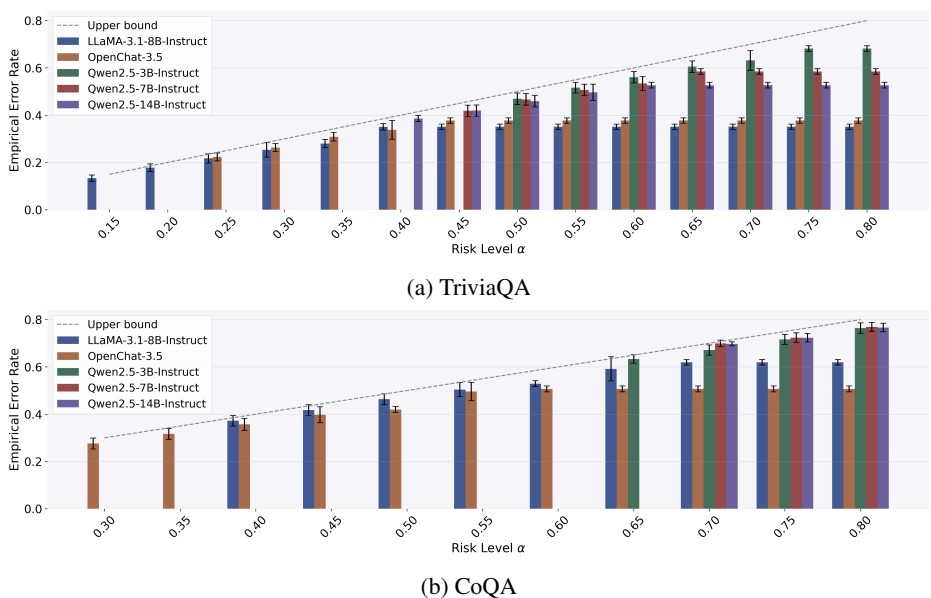

Figure 7: Test-time EER results in the sampling stage with Entailment score as the correctness metric.

Table 2: Test-time EER results in the filtering stage uitilize Rouge-L score as correctness evaluation on the TriviaQA ($\alpha = 0.5$) and CoQA ($\alpha = 0.5$) dataset at various risk levels ($\beta$).

(a) TriviaQA ($\alpha = 0.5$)

| $\beta$ | 0.05 | 0.10 | 0.15 | 0.20 | 0.25 | 0.30 | 0.35 | 0.40 |
|---|---|---|---|---|---|---|---|---|
| Upper Bound ($\alpha + \beta - \alpha\beta$) | 0.525 | 0.55 | 0.575 | 0.6 | 0.625 | 0.65 | 0.675 | 0.7 |
| LLaMA-3.1-8B-Instruct | 0.1491±0.0209 | 0.2466±0.0168 | 0.3182±0.0206 | 0.3855±0.0234 | 0.4413±0.0232 | 0.4872±0.0228 | 0.5340±0.0241 | 0.5789±0.0232 |
| Openchat-3.5 | 0.0569±0.0106 | 0.0961±0.0135 | 0.1385±0.0177 | 0.2001±0.0186 | 0.2459±0.0196 | 0.2966±0.0212 | 0.3459±0.0188 | 0.3891±0.0192 |

(b) CoQA ($\alpha = 0.5$)

| $\beta$ | 0.05 | 0.10 | 0.15 | 0.20 | 0.25 | 0.30 | 0.35 | 0.40 |
|---|---|---|---|---|---|---|---|---|
| Upper Bound ($\alpha + \beta - \alpha\beta$) | 0.525 | 0.55 | 0.575 | 0.6 | 0.625 | 0.65 | 0.675 | 0.7 |
| LLaMA-3.1-8B-Instruct | 0.1314±0.0157 | 0.2072±0.0194 | 0.2750±0.0215 | 0.3289±0.0231 | 0.3809±0.0267 | 0.4321±0.0267 | 0.4845±0.0253 | 0.5341±0.0254 |
| Openchat-3.5 | 0.1345±0.0113 | 0.1681±0.0126 | 0.2053±0.0146 | 0.2432±0.0191 | 0.2836±0.0189 | 0.3208±0.0192 | 0.3588±0.0189 | 0.3936±0.0206 |

**LLM-Based Semantic Evaluation** Following (Zhang et al., 2024), We further examine an LLM-based semantic evaluation criterion, where admissibility is determined by semantic validation from Qwen-2.5-3B-Instruct. This criterion is more flexible than similarity or entailment, as it leverages the model's broader reasoning ability to align outputs with human-like semantic evaluation. Figure 8 reports the sampling stage results on TriviaQA and CoQA. Across all models and risk levels, the empirical error rate remains strictly below the upper bound $\alpha$. Table 4 summarizes the filtering stage results on the same datasets. For all datasets and models, the empirical error rates remain well below the theoretical upper bound $\alpha + \beta - \alpha\beta$ at every setting, and track the bound closely with a small, stable margin. Overall, these findings demonstrate that SAFER retains rigorous two-stage risk control even under an LLM-based semantic evaluation criterion, highlighting its adaptability to human-aligned correctness definitions.

## E.2 EXPERIMENTAL RESULTS ON SCIENCEQA

This section supplements the experimental results and analysis across the sampling stage and filtering stage on ScienceQA under four correctness evaluation criteria: similarity-based matching,

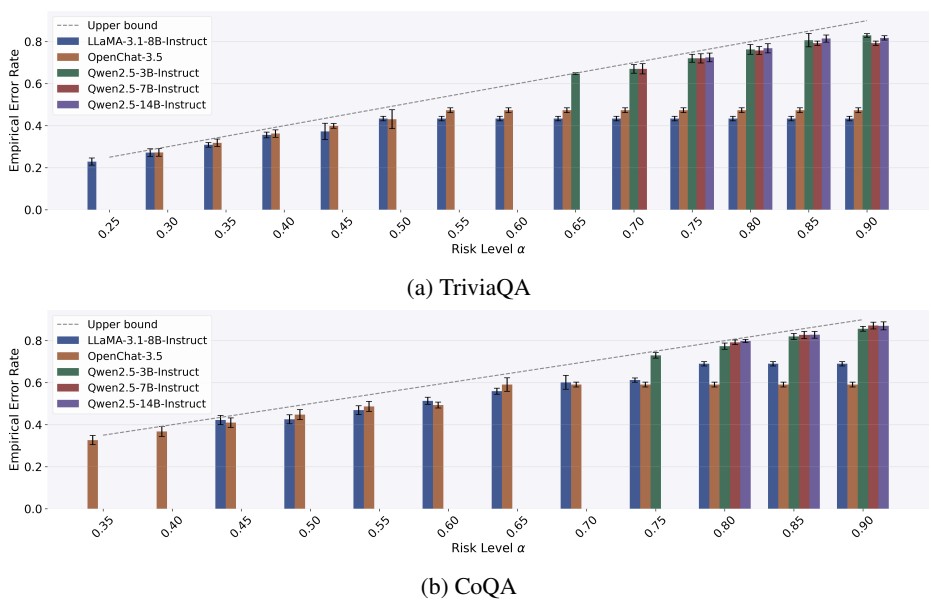

Figure 8: Test-time EER results in the sampling stage with LLM-based semantic evaluation as the correctness metric.

Table 3: Test-time EER results in the filtering stage uitilize entailment as correctness evaluation on the TriviaQA ($\alpha = 0.5$) and CoQA ($\alpha = 0.5$) dataset at various risk levels ($\beta$).

(a) TriviaQA ($\alpha = 0.5$)

| $\beta$ | 0.05 | 0.10 | 0.15 | 0.20 | 0.25 | 0.30 | 0.35 | 0.40 |
|---|---|---|---|---|---|---|---|---|
| Upper Bound ($\alpha + \beta - \alpha\beta$) | 0.525 | 0.55 | 0.575 | 0.6 | 0.625 | 0.65 | 0.675 | 0.7 |
| LLaMA-3.1-8B-Instruct | 0.1782±0.0169 | 0.2752±0.0172 | 0.3422±0.0189 | 0.4059±0.0215 | 0.4589±0.0204 | 0.5022±0.0205 | 0.5459±0.0220 | 0.5858±0.0208 |
| Openchat-3.5 | 0.0936±0.0102 | 0.1339±0.0138 | 0.1752±0.0166 | 0.2288±0.0182 | 0.2771±0.0198 | 0.3274±0.0199 | 0.3730±0.0188 | 0.4157±0.0212 |
| Qwen2.5-3B-Instruct | 0.0506±0.0125 | 0.0994±0.0162 | 0.1431±0.0176 | 0.1891±0.0207 | 0.2330±0.0218 | 0.2774±0.0227 | 0.3252±0.0263 | 0.3829±0.0305 |
| Qwen2.5-7B-Instruct | 0.0475±0.0094 | 0.0894±0.0167 | 0.1403±0.0153 | 0.1826±0.0174 | 0.2288±0.0213 | 0.2776±0.0210 | 0.3252±0.0240 | 0.3732±0.0263 |
| Qwen2.5-14B-Instruct | 0.0646±0.0127 | 0.1219±0.0165 | 0.1671±0.0175 | 0.2128±0.0203 | 0.2597±0.0207 | 0.3015±0.0210 | 0.3425±0.0226 | 0.3873±0.0214 |

(b) CoQA ($\alpha = 0.5$)

| $\beta$ | 0.05 | 0.10 | 0.15 | 0.20 | 0.25 | 0.30 | 0.35 | 0.40 |
|---|---|---|---|---|---|---|---|---|
| Upper Bound ($\alpha + \beta - \alpha\beta$) | 0.525 | 0.55 | 0.575 | 0.6 | 0.625 | 0.65 | 0.675 | 0.7 |
| LLaMA-3.1-8B-Instruct | 0.1699±0.0152 | 0.2356±0.0203 | 0.2926±0.0203 | 0.3408±0.0217 | 0.3865±0.2395 | 0.4333±0.0149 | 0.4761±0.0276 | 0.5250±0.0298 |
| Openchat-3.5 | 0.1719±0.0109 | 0.2050±0.0138 | 0.2391±0.0169 | 0.2755±0.0181 | 0.3102±0.0189 | 0.3486±0.0203 | 0.3826±0.0229 | 0.3826±0.0229 |

Table 4: Test-time EER results in the filtering stage uitilize LLM as correctness evaluation on the TriviaQA ($\alpha = 0.5$) and CoQA ($\alpha = 0.5$) dataset at various risk levels ($\beta$).

(a) TriviaQA ($\alpha = 0.5$)

| $\beta$ | 0.05 | 0.10 | 0.15 | 0.20 | 0.25 | 0.30 | 0.35 | 0.40 |
|---|---|---|---|---|---|---|---|---|
| Upper Bound ($\alpha + \beta - \alpha\beta$) | 0.525 | 0.55 | 0.575 | 0.6 | 0.625 | 0.65 | 0.675 | 0.7 |
| LLaMA-3.1-8B-Instruct | 0.1732±0.0224 | 0.2869±0.0219 | 0.3681±0.0236 | 0.4323±0.0232 | 0.4862±0.0243 | 0.5329±0.0237 | 0.5754±0.0223 | 0.6089±0.0243 |
| Openchat-3.5 | 0.0481±0.0111 | 0.0842±0.0122 | 0.1233±0.0151 | 0.1718±0.0192 | 0.2205±0.0214 | 0.2677±0.0203 | 0.3167±0.0223 | 0.3621±0.0241 |

(b) CoQA ($\alpha = 0.5$)

| $\beta$ | 0.05 | 0.10 | 0.15 | 0.20 | 0.25 | 0.30 | 0.35 | 0.40 |
|---|---|---|---|---|---|---|---|---|
| Upper Bound ($\alpha + \beta - \alpha\beta$) | 0.525 | 0.55 | 0.575 | 0.6 | 0.625 | 0.65 | 0.675 | 0.7 |
| LLaMA-3.1-8B-Instruct | 0.1472±0.0143 | 0.2103±0.0162 | 0.2653±0.0193 | 0.3097±0.0228 | 0.3575±0.0241 | 0.4033±0.0244 | 0.4477±0.0257 | 0.4934±0.0237 |
| Openchat-3.5 | 0.1552±0.0128 | 0.1879±0.0151 | 0.2221±0.0174 | 0.2546±0.0185 | 0.2887±0.0197 | 0.3246±0.0214 | 0.3664±0.0227 | 0.4081±0.0251 |

ROUGE-L score, bi-entailment based on natural language inference (NLI), and LLM-based semantic evaluation.

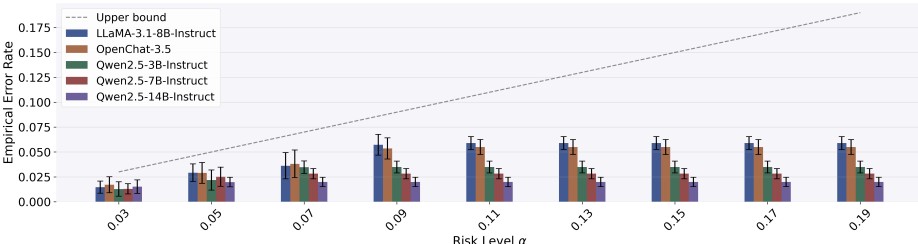

Figure 9: Test-time EER results in the sampling stage with similarity score as the correctness metric on ScienceQA.

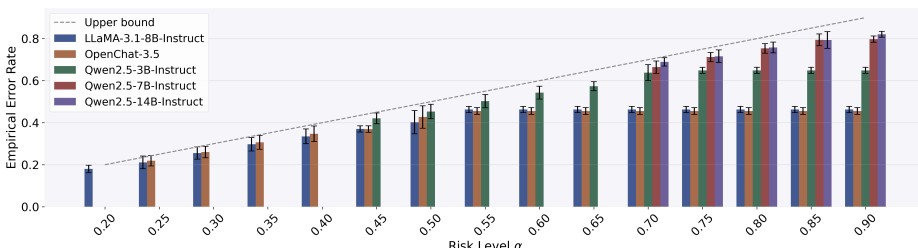

Figure 10: Test-time EER results in the sampling stage with Rouge-L score as the correctness metric on ScienceQA.

Under similarity-based evaluation, we follow the same experimental setup described previously and adopt a threshold of 0.6 to determine correctness on the ScienceQA dataset. We first calibrate the minimal sampling budget on the calibration split for each target risk level $\alpha$ and then assess on the test split. As shown in Figure 9, the test-time EER stays strictly below $\alpha$ for all models and all risk levels in the sampling stage. In the filtering stage, Table 5 reports that for every $\beta$ the final EER is controlled below $\alpha + \beta - \alpha\beta$.

Using ROUGE-L score as the correctness criterion and set a threshold of 0.3, the same calibration protocol ensures that, for every $\alpha$, test-time EER remains strictly below upper bound $\alpha$ for all models in the sampling stage, as shown in Figure 10. In the filtering stage, when varying the risk level $\beta$, the final EER remains below the theoretical bound $\alpha + \beta - \alpha\beta$ for every $\beta$, while the prediction set size decreases monotonically as $\beta$ increases, as reported in Table 6.

For bi-entailment as correctness evaluation, we follow the same two-stage protocol described earlier and set a threshold of 0.5, it is shown in Figure 11 in the sampling stage that across models and target levels, the test-time EER remains strictly below $\alpha$. In the filtering stage, varying the risk level $\beta$ keeps the final EER within the theoretical bound $\alpha + \beta - \alpha\beta$ for every $\beta$ in Table 7.

Under LLM-based correctness evaluation on ScienceQA, we adopt Qwen2.5-3B-Instruct as the judging model and follow the same experimental setup as in previous sections. The results are shown in Figure 12 and Table 8. In the sampling stage, EER on the test split remains strictly below the target risk level $\alpha$ across all models, confirming that the abstention-aware calibration procedure preserves valid control. Compared with similarity and Rouge-L metrics, LLM judgement produces more semantically flexible assessments of correctness. In the filtering stage, varying the risk level $\beta$ demonstrates consistent behavior with theoretical expectations. As reported in Table 8, for each $\beta$ the final EER is strictly bounded by $\alpha + \beta - \alpha\beta$. Among the evaluated models, Qwen2.5-3B-Instruct as the judge yields the strongest calibration efficiency, resulting in smaller error rates. Across different correctness evaluations, SAFER provides uniform, model-agnostic risk control, meeting the corresponding upper bound across the sampling and filtering stages, demonstrating its robust abstention-aware calibration.

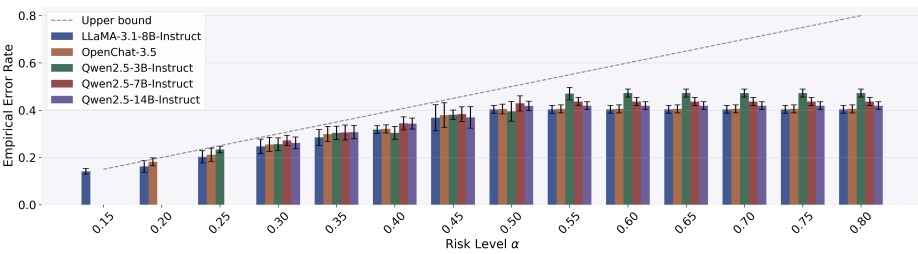

Figure 11: Test-time EER results in the sampling stage with entailment score as the correctness metric on ScienceQA.

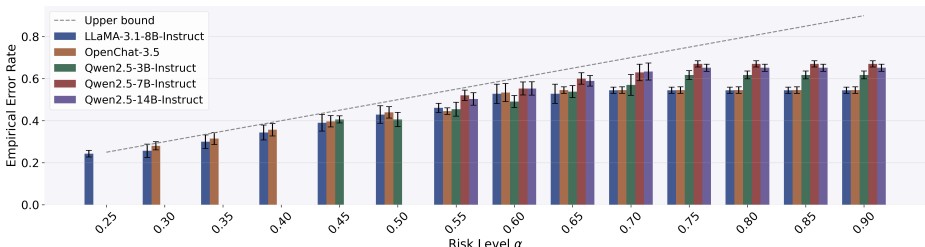

Figure 12: Test-time EER results in the sampling stage with LLM-based semantic evaluation as the correctness metric on ScienceQA.

Table 5: Test-time EER results in the filtering stage uitilize similarity as correctness evaluation on the ScienceQA ($\alpha = 0.05$) dataset at various risk levels ($\beta$).

| $\beta$ | 0.05 | 0.10 | 0.15 | 0.20 | 0.25 | 0.30 | 0.35 | 0.40 |
|---|---|---|---|---|---|---|---|---|
| Upper Bound ($\alpha + \beta - \alpha\beta$) | 0.525 | 0.55 | 0.575 | 0.6 | 0.625 | 0.65 | 0.675 | 0.7 |
| LLaMA-3.1-8B-Instruct | 0.0768±0.0162 | 0.1260±0.0215 | 0.1734±0.0252 | 0.2191±0.0273 | 0.2656±0.0289 | 0.3155±0.0318 | 0.3639±0.0337 | 0.4110±0.0341 |
| Openchat-3.5 | 0.0767±0.0163 | 0.1248±0.0205 | 0.1718±0.0234 | 0.2193±0.0267 | 0.2660±0.0298 | 0.3122±0.0302 | 0.3610±0.0330 | 0.4079±0.0350 |
| Qwen2.5-3B-Instruct | 0.0685±0.0185 | 0.1179±0.0244 | 0.1657±0.0264 | 0.2124±0.0271 | 0.2621±0.0300 | 0.3078±0.0329 | 0.3565±0.0324 | 0.4058±0.0344 |
| Qwen2.5-7B-Instruct | 0.0701±0.0178 | 0.1220±0.0220 | 0.1718±0.0246 | 0.2216±0.0278 | 0.2686±0.0299 | 0.3154±0.0295 | 0.3592±0.0303 | 0.4078±0.0297 |
| Qwen2.5-14B-Instruct | 0.0672±0.0145 | 0.1163±0.0217 | 0.1659±0.0265 | 0.2140±0.0278 | 0.2615±0.0300 | 0.3085±0.0336 | 0.3555±0.0329 | 0.4044±0.0381 |

Table 6: Test-time EER results in the filtering stage uitilize Rouge-L score as correctness evaluation on the ScienceQA ($\alpha = 0.5$) dataset at various risk levels ($\beta$).

| $\beta$ | 0.05 | 0.10 | 0.15 | 0.20 | 0.25 | 0.30 | 0.35 | 0.40 |
|---|---|---|---|---|---|---|---|---|
| Upper Bound ($\alpha + \beta - \alpha\beta$) | 0.525 | 0.55 | 0.575 | 0.6 | 0.625 | 0.65 | 0.675 | 0.7 |
| LLaMA-3.1-8B-Instruct | 0.0966±0.0241 | 0.1772±0.0399 | 0.2522±0.0478 | 0.3170±0.0512 | 0.3767±0.0450 | 0.4354±0.0463 | 0.4876±0.0487 | 0.5343±0.0477 |
| Openchat-3.5 | 0.0481±0.0148 | 0.0909±0.0218 | 0.1409±0.0264 | 0.1852±0.0290 | 0.2294±0.0307 | 0.2723±0.0290 | 0.3173±0.0319 | 0.3625±0.0328 |
| Qwen2.5-3B-Instruct | 0.0284±0.0128 | 0.0603±0.0169 | 0.0901±0.0189 | 0.1239±0.0234 | 0.1576±0.0250 | 0.1908±0.0287 | 0.2252±0.0293 | 0.2638±0.0327 |

Table 7: Test-time EER results in the filtering stage uitilize entailment as correctness evaluation on the ScienceQA ($\alpha = 0.5$) dataset at various risk levels ($\beta$).

| $\beta$ | 0.05 | 0.10 | 0.15 | 0.20 | 0.25 | 0.30 | 0.35 | 0.40 |
|---|---|---|---|---|---|---|---|---|
| Upper Bound ($\alpha + \beta - \alpha\beta$) | 0.525 | 0.55 | 0.575 | 0.6 | 0.625 | 0.65 | 0.675 | 0.7 |
| LLaMA-3.1-8B-Instruct | 0.1303±0.0267 | 0.2168±0.0279 | 0.2820±0.0287 | 0.3371±0.0322 | 0.3922±0.0336 | 0.4511±0.0330 | 0.5013±0.0364 | 0.5545±0.0357 |
| Openchat-3.5 | 0.0721±0.0154 | 0.1088±0.0185 | 0.1505±0.0263 | 0.1992±0.0290 | 0.2446±0.0271 | 0.2855±0.0258 | 0.3262±0.0273 | 0.3721±0.0330 |
| Qwen2.5-3B-Instruct | 0.0417±0.0134 | 0.0786±0.0184 | 0.1125±0.0227 | 0.1498±0.0255 | 0.1873±0.0281 | 0.2220±0.0273 | 0.2613±0.0310 | 0.3022±0.0346 |
| Qwen2.5-7B-Instruct | 0.0373±0.0127 | 0.0718±0.0183 | 0.1063±0.0216 | 0.1446±0.0268 | 0.1870±0.0301 | 0.2247±0.0319 | 0.2644±0.0328 | 0.3071±0.0340 |
| Qwen2.5-14B-Instruct | 0.0317±0.0115 | 0.0618±0.0163 | 0.0957±0.0237 | 0.1337±0.0266 | 0.1756±0.0315 | 0.2190±0.0331 | 0.2662±0.0367 | 0.3111±0.0348 |

## E.3 VALIDATION ON BLACK-BOX MODELS

Crucially, the choice of uncertainty heuristic affects only the efficiency (prediction set size) rather than the validity (risk control guarantee), as the conformal calibration remains distribution-free. A

Table 8: Test-time EER results in the filtering stage uitilize LLM as correctness evaluation on the ScienceQA ($\alpha = 0.5$) dataset at various risk levels ($\beta$).

| $\beta$ | 0.05 | 0.10 | 0.15 | 0.20 | 0.25 | 0.30 | 0.35 | 0.40 |
|---|---|---|---|---|---|---|---|---|
| Upper Bound ($\alpha + \beta - \alpha\beta$) | 0.525 | 0.55 | 0.575 | 0.6 | 0.625 | 0.65 | 0.675 | 0.7 |
| LLaMA-3.1-8B-Instruct | 0.1054±0.0240 | 0.1876±0.0321 | 0.2760±0.0422 | 0.3445±0.0380 | 0.3984±0.0380 | 0.4469±0.0393 | 0.4951±0.0386 | 0.5341±0.0362 |
| Openchat-3.5 | 0.0476±0.0117 | 0.0850±0.0218 | 0.1294±0.0254 | 0.1719±0.0245 | 0.2124±0.0257 | 0.2507±0.0319 | 0.2955±0.0321 | 0.3388±0.0329 |
| Qwen2.5-3B-Instruct | 0.0375±0.0138 | 0.0706±0.0171 | 0.1014±0.0197 | 0.1375±0.0221 | 0.1708±0.0243 | 0.2091±0.0289 | 0.2515±0.0308 | 0.2932±0.0313 |

stronger heuristic yields smaller prediction sets, while a weaker one results in larger sets to satisfy the same coverage requirement.

To demonstrate the applicability of SAFER to black-box models where internal logits are inaccessible, we evaluate GPT-4o-mini on the CoQA dataset. We employ a consistency-based frequency score as the uncertainty heuristic $U(\cdot)$, defined as $1 - \text{Frequency}(\hat{y})$, where frequency denotes the proportion of semantically equivalent answers within the sampled set.

Table 9 and Table 10 present the empirical results for Stage I and Stage II, respectively. In the sampling stage, the empirical miscoverage rate consistently remains below the target risk level $\alpha$. In the filtering stage, with a fixed $\alpha = 0.25$, the final empirical risk is strictly bounded by the theoretical limit ($\alpha + \beta - \alpha\beta$) across all total risk budgets. These results confirm that SAFER maintains rigorous statistical guarantees even in black-box settings.

Table 9: Empirical risk (Mean ± Std) of Stage I on CoQA using GPT-4o-mini across varying feasibility risk levels ($\alpha$).

| $\alpha$ | Empirical Risk |
|---|---|
| 0.25 | $0.2293 \pm 0.0173$ |
| 0.30 | $0.2462 \pm 0.0229$ |
| 0.35 | $0.2745 \pm 0.0234$ |
| 0.40 | $0.2934 \pm 0.0110$ |
| 0.45 | $0.2960 \pm 0.0036$ |
| 0.50 | $0.2960 \pm 0.0036$ |
| 0.55 | $0.2960 \pm 0.0036$ |

Table 10: Empirical risk (Mean ± Std) of Stage II on CoQA using GPT-4o-mini with consistency-based frequency scores. Experiments use fixed $\alpha = 0.25$ and varying total risk budgets.

| $\alpha + \beta - \alpha\beta$ | Empirical Risk |
|---|---|
| 0.2875 | $0.2293 \pm 0.0173$ |
| 0.3250 | $0.2293 \pm 0.0173$ |
| 0.3625 | $0.2293 \pm 0.0173$ |
| 0.4000 | $0.2293 \pm 0.0173$ |
| 0.4375 | $0.2674 \pm 0.0834$ |
| 0.4750 | $0.3897 \pm 0.1116$ |
| 0.5125 | $0.4691 \pm 0.0852$ |
| 0.5500 | $0.5244 \pm 0.0376$ |
| 0.5875 | $0.5570 \pm 0.0323$ |
| 0.6250 | $0.5913 \pm 0.0388$ |

### E.4 SENSITIVITY ANALYSIS ON SAMPLING BUDGET

The maximum sampling cap $M$ serves as a critical hyperparameter governing the trade-off between inference cost and the system's ability to uncover admissible answers. We analyze the impact of $M$ on the abstention rate using the CoQA dataset, comparing two models of different sizes: Qwen2.5-14B-Instruct and Qwen2.5-7B-Instruct.

As presented in Figure 13, increasing $M$ significantly reduces the abstention rate for both models, demonstrating the utility of expanding the search space. For instance, with Qwen2.5-7B, the rate drops from 37.22% at $M = 1$ to roughly 22.3% at $M = 19$. However, this benefit exhibits diminishing returns, saturating as $M$ approaches 20.

Furthermore, the results highlight the impact of model capability: the larger 14B model consistently achieves lower abstention rates compared to the 7B model under the same budget. This indicates that while increasing the sampling budget can compensate for model limitations to an extent, a stronger underlying model fundamentally improves the feasibility of risk control.

### E.5 GENERALIZATION TO MULTIMODAL REASONING

To assess the framework's generalization capabilities beyond text-only tasks, we extended our evaluation to MMVet (Yu et al., 2023), a challenging multi-modal benchmark that requires integrated visual recognition and reasoning capabilities. We evaluated two state-of-the-art vision-language models: Qwen2-VL-7B-Instruct (Wang et al., 2024b) and InternVL2-8B (Chen et al., 2024).

The results confirm that SAFER maintains strict statistical validity in multi-modal settings. Table 11 presents the results for Stage I (Sampling), where the empirical risk consistently tracks the target risk levels ($\alpha$). Table 12 shows the results for Stage II (Filtering) using bi-entailment as the correctness metric (with fixed Stage I risk $\alpha = 0.1$). Across all target risk settings, the final empirical risk remains strictly bounded by the theoretical limit ($\alpha + \beta - \alpha\beta$).

Table 11: Stage I results on MMVet. The empirical risk (Mean $\pm$ Std) for both Qwen2-VL-7B and InternVL2-8B consistently satisfies the user-specified risk level ($\alpha$).

| Risk Level ($\alpha$) | Empirical Risk (Qwen2-VL-7B) | Empirical Riskv (InternVL2-8B) |
| --- | --- | --- |
| 0.10 | $0.0819 \pm 0.0275$ | $0.0917 \pm 0.0153$ |
| 0.15 | $0.0835 \pm 0.0376$ | $0.0901 \pm 0.0337$ |
| 0.20 | $0.1182 \pm 0.0398$ | $0.1252 \pm 0.0375$ |
| 0.25 | $0.1438 \pm 0.0337$ | $0.1582 \pm 0.0453$ |
| 0.30 | $0.1484 \pm 0.0259$ | $0.1882 \pm 0.0400$ |
| 0.35 | $0.1484 \pm 0.0259$ | $0.1955 \pm 0.0275$ |
| 0.40 | $0.1484 \pm 0.0259$ | $0.1955 \pm 0.0275$ |
| 0.45 | $0.1484 \pm 0.0259$ | $0.1955 \pm 0.0275$ |

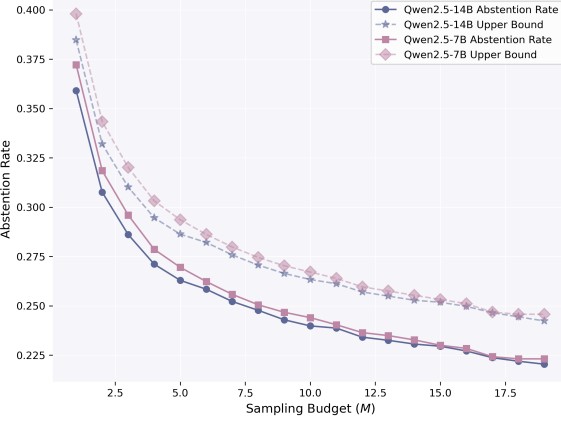

Figure 13: Abstention rates and Clopper-Pearson upper bounds on the CoQA dataset for Qwen2.5-14B and Qwen2.5-7B across varying sampling budgets ($M$).

Table 12: Stage II results on MMVet using bi-entailment for correctness evaluation. With a fixed Stage I risk $\alpha = 0.1$, the final empirical risk stays below the theoretical upper bound $(\alpha + \beta - \alpha\beta)$ across varying total risk levels.

| Risk Level $(\alpha + \beta - \alpha\beta)$ | Empirical Risk (Qwen2-VL) | Empirical Risk (InternVL2-8B) |
|---|---|---|
| 0.145 | $0.114 \pm 0.031$ | $0.113 \pm 0.025$ |
| 0.190 | $0.152 \pm 0.041$ | $0.153 \pm 0.040$ |
| 0.235 | $0.190 \pm 0.045$ | $0.182 \pm 0.046$ |
| 0.280 | $0.239 \pm 0.051$ | $0.236 \pm 0.046$ |
| 0.325 | $0.287 \pm 0.058$ | $0.273 \pm 0.057$ |
| 0.370 | $0.334 \pm 0.052$ | $0.327 \pm 0.055$ |

---

**TriviaQA**

### System: This is a bot that correctly answers questions.

### User: In 1968, who did radical feminist Valerie Solanas shoot and wound as he entered his New York studio?
### Assistant: Andy Warhol

### User: What lake can be found on the border of Vermont and New York?
### Assistant: Lake Champlain

### User: Which competition was won by Nadiya Hussain in 2015?
### Assistant: The Great British Bake-Off

### User: Who was the man behind The Chipmunks?
### Assistant:

---

Figure 14: A prompt example in the TriviaQA Dataset.

---

**CoQA**

### System: This is a bot that correctly answers questions. Once upon a time, in a barn near a farm house, there lived a little white kitten named Cotton. Cotton lived high up in a nice warm place above the barn where all of the farmer's horses slept. But Cotton wasn't alone in her little home above the barn, oh no. She shared her hay bed with her mommy and 5 other sisters. All of her sisters were cute and fluffy, like Cotton. But she was the only white one in the bunch. The rest of her sisters were all orange with beautiful white tiger stripes like Cotton's mommy. Being different made Cotton quite sad. She often wished she looked like the rest of her family. So one day, when Cotton found a can of the old farmer's orange paint, she used it to paint herself like them. When her mommy and sisters found her they started laughing. "What are you doing, Cotton?!" "I only wanted to be more like you". Cotton's mommy rubbed her face on Cotton's and said "Oh Cotton, but your fur is so pretty and special, like you. We would never want you to be any other way". And with that, Cotton's mommy picked her up and dropped her into a big bucket of water. When Cotton came out she was herself again. Her sisters licked her face until Cotton's fur was all all dry. "Don't ever do that again, Cotton!" they all cried. "Next time you might mess up that pretty white fur of yours and we wouldn't want that!" Then Cotton thought, "I change my mind. I like being special".

### User: What color was Cotton?
### Assistant: white

### User: Where did she live?
### Assistant:

---

Figure 15: A prompt example in the CoQA Dataset.

---

**ScienceQA**

### System: This is a bot that correctly answers questions.

### User: What are the cell walls of fungi made of?
### Assistant: chitin

### User: What is the pattern of spacing among individuals within the boundaries of the population?
### Assistant: dispersion

### User: Fragmentation with subsequent regeneration is a method of what, exhibited by animals such as sea stars?
### Assistant: asexual reproduction

### User: Who proposed the theory of evolution by natural selection?
### Assistant:

---

Figure 16: A prompt example in the ScienceQA Dataset.

---

**MMVet**

<image>
What is $x$ in the equation?
NOTE: Provide only the final answer. Do not provide unrelated details.

---

Figure 17: A prompt example in the MMVet Dataset.

