# OpenReview forum: "SAFER: Risk-Constrained Sample-then-Filter in Large Language Models"
_ICLR.cc/2026/Conference — ICLR 2026 Poster_

### Official Review · Reviewer_z2a5 · 2025-10-24

**Soundness:** 3
**Presentation:** 2
**Contribution:** 2
**Rating:** 2
**Confidence:** 2

**Summary:**

The paper proposers Sampling and conformalized filtering  that use conformal risk control into two stage framework. The first stage calibrates sampling budget for responses candidate set to contain admissible answers at miscoverage rate $\alpha$. The second stage introduces second risk level $\beta$ to control probability of incorrectly filtering out of all admissible responses to produce a prediction set. The final prediction set fails to cover an admissible answer with probability $1-delta$ over the calibration set with bound $\alpha + \beta - \alpha\beta$. Experiments on three QA datasets (TriviaQA, CoQA, ScienceQA) using five different LLMs demonstrate that SAFER validly controls the empirical error rate below the specified bounds.

**Strengths:**

* The abstention aware sampling is mechanism is the main novelty. Should clarify and compare with Yadkori 2024. Is this just an application/extension of existing works?
* The bound on the sampling risk looks to be formally derived and empirically validated across several LLMs and datasets.
* The evaluation of different correctness criteria (sentence similarity, Rouge-L, NLI-based bi-entailment, and LLM-based semantic evaluation) add empirical support

**Weaknesses:**

* The paper is incremental in terms of technical contribution, combining existing LTT and conformal ideas with abstention-aware calibration.
* The calibration for Stage 1 requires generating/evaluating $N\timesM$ samples and evaluating these responses against the ground truth could be very computationally expensive.
* The maximum sampling cap M is a key hyperparameter that is not explored in detail.
* Although many related CP works are cited, the experiments ignores them completely and only compares against single TRON baseline.
* Only QA datasets are used, no evaluation on reasoning, summarization, or multi-modal tasks

**Questions:**

* Is stage I just straightfoward application of LTT and stage II just application of Conformal Risk Control? Can you clarify core contributions beyond just direct application of existing frameworks?
* Is there bias in sequential calibration when stage II calibration is only performed on biased subset of admissible samples from stage I? This seems like form of conditional coverage and is not discussed. How do you justify that this biased selection step does not break exchangeability?
* How to choose $M$ and why are ablation experiments not conducted on such a critical hyperparameter?
* How to extend framework to black-box API-only models without logit access?
* How does the quality of the uncertainty heuristic $U(u)$ affect the efficiency of the final prediction set size?
* Can the 3D plots be visualized in a better way?
* Can the sampling budget be made adaptive to improve efficiency based on example difficulty or some other criteria?
* Can the two-stage  $\alpha + \beta - \alpha\beta$ be made tighter?
* Could the calibration of $\alpha$ and $\beta$ be done jointly rather than separately?
* How does Clopper-Pearson (classical statistical tool) compare to alternative calibration approaches to find confidence intervals?
* How does this framework extend to when admissibility is fuzzy or difficult to measure exactly?
* The framework requires the user to specify two risk levels, how should a user practically choose $\alpha$ and $\beta$ to target a specific overall risk level $\epsilon$ between sampling and filtering?

---

> ### Author Response · Authors · 2025-11-20
> **Responses to Reviewer z2a5 comments (Part 1)**
>
> > **W1:** The paper is incremental in terms of technical contribution, combining existing LTT and conformal ideas with abstention-aware calibration.
>
> ***Response:*** Our contributions center on three components:
>
> (a) determining whether a user-specified risk level is fundamentally feasible under finite sampling,
> (b) computing the minimum sampling budget required to meet that risk level, and
> (c) filtering unreliable samples using a statistically calibrated uncertainty threshold.
>
> Regarding (a), we introduce a feasibility test using Clopper-Pearson intervals to verify if the target risk is achievable. This differs from prior works like ConU [1] and TRON [2], which apply a fixed sampling budget and implicitly assume a correct answer is always obtainable. Our method explicitly detects when low risk levels are infeasible and abstains, avoiding the invalid claims made by fixed-budget methods in open-ended settings.
>
>  (b) Unlike ConU which relies on a fixed sample size (e.g., 20), SAFER computes the minimum sampling size required to meet a feasible risk level. Once feasibility is established, this adaptive sampling strategy substantially reduces test-time cost and provides a principled way to operationalize LTT in open-ended generation settings.
>
>  Finally, for (c), we employ Conformal Risk Control (CRC) to derive a statistically calibrated threshold for filtering incorrect answers. Unlike standard methods that discard answers based solely on heuristic uncertainty rankings—which often breaks statistical guarantees—our integration of CRC eliminates unreliable generations while strictly maintaining the user-specified coverage guarantee.
>
> > **Q1**: Is stage I just straightfoward application of LTT and stage II just application of Conformal Risk Control? Can you clarify core contributions beyond just direct application of existing frameworks?
>
> ***Response:*** We clarify that our core contribution lies in **identifying specific challenges inherent to open-ended generation** and **tailoring the application of LTT and CRC to address them**—nuances that prior works (e.g., ConU, TRON) overlooked.
>
> **Stage I addresses the practicality gap in open-ended generation following the LTT paradigm.** Previous methods apply fixed sampling budgets and implicitly assume that a correct answer is always obtainable. This is invalid for open-ended tasks where specific risk levels may be infeasible. To address this, we apply LTT to construct a feasibility test using Clopper-Pearson intervals to verify if the target risk is accessible on the calibration set. This allows SAFER to strictly abstain when the risk is unattainable or compute the minimum sampling budget when feasible, ensuring statistical validity where fixed-budget assumptions fail.
>
> **Stage II tackles incorrect answers in finite sampling.** Since uncertainty heuristics are imperfect, we employ CRC to derive a statistically rigorous threshold. This filters unreliable generations while strictly maintaining coverage guarantees, ensuring the prediction set remains valid.
>
> > **W2** The calibration for Stage 1 requires generating/evaluating N\timesM samples and evaluating these responses against the ground truth could be very computationally expensive.
>
> ***Response:*** We may appropriately reduce N. Figure 6 also shows that even when the calibration–test split ratio is 0.1, the method still achieves statistically valid coverage guarantees. In this sense, we trade computational cost for risk control, which is fully consistent with the foundational requirements of split conformal prediction.
>
>
>
> ---
>
> ### References
>
> [1]  "Conu: Conformal uncertainty in large language models with correctness coverage guarantees." Findings of the Association for Computational Linguistics: EMNLP 2024. 2024.
>
> [2]  "Sample then identify: A general framework for risk control and assessment in multimodal large language models." arXiv preprint arXiv:2410.08174 (2024).

---

> ### Author Response · Authors · 2025-11-20
> **Responses to Reviewer z2a5 comments (Part 2)**
>
> > **Q3 and W3:** How to choose M and why are ablation experiments not conducted on such a critical hyperparameter?/The maximum sampling cap M is a key hyperparameter that is not explored in detail.
>
> ***Response:*** $M$ directly determines the lower bound of feasible risk level. Meanwhile, if the predictive capability of the deployed LLM is limited, the lower bound of feasible risk level won’t fall when $M$ reaches a certain high point. We validated this by conducting an experiment on the CoQA dataset using Qwen2.5-7B-Instruct (relevance method: similarity > 0.6). As shown in the table below, the risk upper bound decreases initially but **remains unchanged** at 0.2457 for $M \geq$18,
>
> |  $M$   | Abstention Rate | Upper Bound |
> | :--: | :-------------: | :---------: |
> |  1   |     0.3722      |   0.3980    |
> |  2   |     0.3185      |   0.3434    |
> |  3   |     0.2960      |   0.3201    |
> |  4   |     0.2786      |   0.3032    |
> |  5   |     0.2695      |   0.2936    |
> |  6   |     0.2623      |   0.2862    |
> |  7   |     0.2558      |   0.2798    |
> |  8   |     0.2505      |   0.2745    |
> |  9   |     0.2467      |   0.2703    |
> |  14  |     0.2327      |   0.2553    |
> |  15  |     0.2300      |   0.2532    |
> |  16  |     0.2284      |   0.2511    |
> |  17  |     0.2242      |   0.2468    |
> |  18  |     0.2231      |   0.2457    |
> |  19  |     0.2231      |   0.2457    |
> |  20  |     0.2226      |   0.2457    |
>
>
> > The abstention aware sampling mechanism is the main novelty. Should clarify and compare with Yadkori 2024. Is this just an application/extension of existing works?
>
> ***Response:*** Yadkori et al. (2024) [1] adopt a point-wise abstention mechanism: a single selected answer is scored and a threshold is calibrated. Their formulation does not model the finite-sample possibility that a sampled set may contain no correct answer. Our framework incorporates an explicit estimation of this quantity. In our experiments, this event occurs with non-trivial frequency—for example, with $M=1$ the estimated rate is 0.3722 (upper bound 0.3980), and it decreases only gradually to 0.2226 at $M=20$ (upper bound 0.2457)—which motivates the need for an abstention-aware sampling stage.
>
> > **W4:** Although many related CP works are cited, the experiments ignores them completely and only compares against single TRON baseline.
>
> ***Response:*** ConU [2] and SE-SCP [5] both assume that the calibration and test samples can always obtain at least one correct answer within a finite sampling budget. This assumption is unrealistic, and under our setting these methods do not work. LofreeCP [4] does not enforce a strict alignment between the nonconformity score and correctness during calibration, and thus cannot guarantee coverage at arbitrary user-specified risk levels during testing. We only compare against TRON [3] because it explicitly considers whether the sampling set contains a correct answer; other related methods are not directly comparable. Our work is specifically designed to address the challenge of ensuring correct coverage under finite sampling while simultaneously filtering out incorrect answers from the sampling set without compromising the statistical guarantees.
>
>
> ---
>
> ### References
>
> [1] "Mitigating llm hallucinations via conformal abstention." *arXiv preprint arXiv:2405.01563* (2024).
>
> [2] "Sample then identify: A general framework for risk control and assessment in multimodal large language models." arXiv preprint arXiv:2410.08174 (2024).
>
> [3] "Conu: Conformal uncertainty in large language models with correctness coverage guarantees." Findings of the Association for Computational Linguistics: EMNLP 2024. 2024.
>
> [4] "Api is enough: Conformal prediction for large language models without logit-access." *arXiv preprint arXiv:2403.01216* (2024).
>
> [5] "Addressing uncertainty in llms to enhance reliability in generative ai." *arXiv preprint arXiv:2411.02381* (2024).

---

> ### Author Response · Authors · 2025-11-20
> **Responses to Reviewer z2a5 comments (Part 3)**
>
> > **Q2:** Is there bias in sequential calibration when stage II calibration is only performed on biased subset of admissible samples from stage I? This seems like form of conditional coverage and is not discussed. How do you justify that this biased selection step does not break exchangeability?
>
> ***Response:*** We justify this design as follows:
>
> 1. **Necessity for CRC:** The application of the CRC framework requires calibrating a threshold **aligned with the correct answer** to filter out unreliable answers while **maintaining guarantees**. The loss function in CRC (Eq. 12) is legally defined only when the sampling set contains at least one correct answer. Therefore, restricting calibration to this subset is a prerequisite for optimizing the threshold.
> 2. **Preservation of Exchangeability:** **This selection does not break exchangeability regarding the guarantee.** Actually, the "bias" (selection rule) is applied symmetrically to both calibration and test data. Since the total risk is decomposed (Eq. 22), the Stage I budget alpha accounts for test samples where no answer exists. Consequently, since Stage II specifically targets the conditional risk (Eq. 14), exchangeability is only required (and satisfied) between the calibration subset and the corresponding sub-population of test samples where correct answers exist.
>
>
>
> > **Q4:**How to extend framework to black-box API-only models without logit access?
>
> ***Response：*** Our framework admits any uncertainty methods in the second stage. For the GPT-4o-mini experiments on the CoQA dataset, we utilize "one minus consistency-based frequency score" to compute the uncertainty score of each sampled answer. The correctness of an answer is determined using a threshold criterion: similarity > 0.6. The initial plateau reflects the baseline error from Stage I sampling failures, while the subsequent rise results from the additional miscoverage introduced by Stage II filtering as the risk tolerance β increases. **Essentially, at low risk levels, the filter is conservative and retains most candidates, so the error is determined solely by Stage I; as the risk budget expands, the filter becomes more aggressive in removing answers, which introduces additional filtering errors.**
>
> Stage I:
>
> | $\alpha$ |   Mean $\pm$ Std    |
> | :------: | :-----------------: |
> |   0.25   | 0.2293 $\pm$ 0.0173 |
> |   0.30   | 0.2462 $\pm$ 0.0229 |
> |   0.35   | 0.2745 $\pm$ 0.0234 |
> |   0.40   | 0.2934 $\pm$ 0.0110 |
> |   0.45   | 0.2960 $\pm$ 0.0036 |
> |   0.50   | 0.2960 $\pm$ 0.0036 |
> |   0.55   | 0.2960 $\pm$ 0.0036 |
>
> Stage II: (alpha=0.25)
>
> | $\alpha + \beta - \alpha\beta$ |   Mean $\pm$ Std    |
> | :----------------------------: | :-----------------: |
> |             0.2875             | 0.2293 $\pm$ 0.0173 |
> |             0.3250             | 0.2293 $\pm$ 0.0173 |
> |             0.3625             | 0.2293 $\pm$ 0.0173 |
> |             0.4000             | 0.2293 $\pm$ 0.0173 |
> |             0.4375             | 0.2674 $\pm$ 0.0834 |
> |             0.4750             | 0.3897 $\pm$ 0.1116 |
> |             0.5125             | 0.4691 $\pm$ 0.0852 |
> |             0.5500             | 0.5244 $\pm$ 0.0376 |
> |             0.5875             | 0.5570 $\pm$ 0.0323 |
> |             0.6250             | 0.5913 $\pm$ 0.0388 |
>
>
>
> > **Q5:** How does the quality of the uncertainty heuristic U(u) affect the efficiency of the final prediction set size?
>
>
>
> ***Response:*** The performance of uncertainty methods to separate incorrect from correct answers do not affect the final guarantee, but affect the quality of prediction sets. The weaker the uncertainty method, the larger the average prediction set size.
>
> > **Q6:** Can the 3D plots be visualized in a better way?
>
> ***Response:*** We have updated Figure 4 in the revised PDF to improve readability.
>
>
>
> > **Q7:** Can the sampling budget be made adaptive to improve efficiency based on example difficulty or some other criteria?
>
> ***Response:*** The maximum sampling budget is determined by the user. In black-box scenarios, a larger sampling budget yields more discriminative uncertainty scores, thereby reducing the prediction set size under the same risk level $\beta$. If the risk level is feasible, the test-time minimum sampling budget is computed in the first stage of SAFER.

---

> ### Author Response · Authors · 2025-11-20
> **Responses to Reviewer z2a5 comments (Part 4)**
>
> > **Q8:** Can the two-stage alpha+beta- alpha*beta be made tighter? **Q9:** Could the calibration of alpha and beta be done jointly rather than separately?
>
> ***Response:*** We address these linked questions together. Our current two-stage design intentionally decouples feasibility estimation (Stage I) from uncertainty filtering (Stage II) to ensure interpretability. This separation allows us to strictly distinguish between "infeasible prompts" (where the model must abstain entirely) and "noisy generations" (where the model must filter specific answers). This modularity provides robust, distribution-free guarantees and transparent diagnostics for why the system rejects a response.
>
> We see joint calibration as a promising optimization. A natural first step would be defining a unified non-conformity score that combines feasibility and uncertainty metrics, while keeping separate thresholds for each stage. This would capture the interaction between stages, tighten the coverage bound, and simplify the parameter space, all while learning dependencies directly from the data.
>
>
>
> > **Q10:** How does Clopper-Pearson (classical statistical tool) compare to alternative calibration approaches to find confidence intervals?
>
> ***Response:*** We employ Clopper-Pearson because it provides exact confidence intervals, whereas alternatives like Hoeffding (HF) are overly conservative. As shown in the table below (CoQA, Qwen2.5-14B), Clopper-Pearson consistently requires a smaller sampling budget $\hat{s}$ than HF to meet the same risk, significantly improving data efficiency
>
> | $\alpha$ | CP $\hat{s}$ | HF $\hat{s}$ |
> | :------: | :----------: | :----------: |
> |  0.250   | 13.2$\pm$4.7 | 15.7$\pm$2.1 |
> |  0.300   | 9.8$\pm$5.0  | 12.4$\pm$5.1 |
> |  0.350   | 3.7$\pm$2.5  | 6.4$\pm$3.8  |
> |  0.400   | 1.9$\pm$0.8  | 2.8$\pm$2.1  |
> |  0.450   | 1.1$\pm$0.4  | 1.5$\pm$0.6  |
> |  0.500   | 1.0$\pm$0.1  | 1.1$\pm$0.2  |
>
>
>
> > **Q11:** How does this framework extend to when admissibility is fuzzy or difficult to measure exactly?
>
> ***Response:*** Our framework guarantees validity based on **exchangeability**, not the precision of the admissibility metric. As long as the chosen measure (even if fuzzy or noisy) is consistently applied to both calibration and test data, the statistical guarantees hold strictly with respect to that specific measure.
>
>
>
> > **Q12:** The framework requires the user to specify two risk levels, how should a user practically choose and to target a specific overall risk level between sampling and filtering?
>
> ***Response:*** Practically, $\alpha$ is set to the minimum feasible risk level achievable at the maximum allowable sampling budget. Since $\beta$ governs the quality of the final prediction sets, we typically fix it at 0.1, consistent with previous studies.
>
>
>
> > **W5:** Only QA datasets are used, no evaluation on reasoning, summarization, or multi-modal tasks
>
> ***Response:*** We extended our evaluation to **MMVet** [1] , a complex multi-modal reasoning benchmark, using **Qwen2-VL-7B** [2] and **InternVL2-8B** [3]. As shown below, the empirical risk (Mean) consistently stays within the target bounds across all risk levels. This confirms that SAFER's distribution-free guarantees generalize effectively to multi-modal reasoning tasks beyond standard QA.
>
>  Table 1: Stage I Results on MMVet
> | Risk Level ($\alpha$) | Qwen2-VL-7B (Mean $\pm$ Std) | InternVL2-8B (Mean $\pm$ Std) |
> | :---: | :---: | :---: |
> | 0.10 | 0.0819 $\pm$ 0.0275 | 0.0917 $\pm$ 0.0153 |
> | 0.15 | 0.0835 $\pm$ 0.0376 | 0.0901 $\pm$ 0.0337 |
> | 0.20 | 0.1182 $\pm$ 0.0398 | 0.1252 $\pm$ 0.0375 |
> | 0.25 | 0.1438 $\pm$ 0.0337 | 0.1582 $\pm$ 0.0453 |
> | 0.30 | 0.1484 $\pm$ 0.0259 | 0.1882 $\pm$ 0.0400 |
> | 0.35 | 0.1484 $\pm$ 0.0259 | 0.1955 $\pm$ 0.0275 |
> | 0.40 | 0.1484 $\pm$ 0.0259 | 0.1955 $\pm$ 0.0275 |
> | 0.45 | 0.1484 $\pm$ 0.0259 | 0.1955 $\pm$ 0.0275 |
>
> Table 2: Stage II Results on MMVet (Correctness: Bi-entailment, $\alpha$=0.1)
> | Risk Level ($\alpha+\beta-\alpha\beta$) | Qwen2-VL (Mean$\pm$Std) | InternVL2 (Mean$\pm$Std) |
> | :---: | :---: | :---: |
> | 0.145 | 0.114 $\pm$ 0.031 | 0.113 $\pm$ 0.025 |
> | 0.190 | 0.152 $\pm$ 0.041 | 0.153 $\pm$ 0.040 |
> | 0.235 | 0.190 $\pm$ 0.045 | 0.182 $\pm$ 0.046 |
> | 0.280 | 0.239 $\pm$ 0.051 | 0.236 $\pm$ 0.046 |
> | 0.325 | 0.287 $\pm$ 0.058 | 0.273 $\pm$ 0.057 |
> | 0.370 | 0.334 $\pm$ 0.052 | 0.327 $\pm$ 0.055 |
>
>
>
> ---
>
> ### References
>
> [1] Mm-vet: Evaluating large multimodal models for integrated capabilities[J]. arXiv preprint arXiv:2308.02490, 2023.
>
> [2] Qwen2-vl: Enhancing vision-language model's perception of the world at any resolution[J]. arXiv preprint arXiv:2409.12191, 2024.
>
> [3] Internvl: Scaling up vision foundation models and aligning for generic visual-linguistic tasks[C]//Proceedings of the IEEE/CVF conference on computer vision and pattern recognition. 2024: 24185-24198.

---

> ### Author Response · Authors · 2025-11-26
> **Have we addressed your concern?**
>
> Dear Reviewer z2a5,
>
> Thank you again for taking the time to review our paper and providing detailed feedback. As the end of the discussion period is approaching, we want to follow up to see if you have any additional questions we have not addressed, or if there is anything else we can help clarify. We have tried responding to the comments from your initial review, and we are more than happy to discuss any points further. Thank you!
>
> Authors of paper 3550

---

### Official Review · Reviewer_mzfb · 2025-10-29

**Soundness:** 3
**Presentation:** 2
**Contribution:** 2
**Rating:** 6
**Confidence:** 4

**Summary:**

This paper introduces SAFER, a two-stage framework for uncertainty quantification in open-ended question answering with LLMs. The framework consists of: (1) abstention-aware sampling that calibrates a minimum sampling budget using the Clopper-Pearson exact method, abstaining when risk constraints cannot be satisfied, and (2) conformalized filtering that removes unreliable candidates using conformal risk control. The authors provide theoretical guarantees that miscoverage probability is bounded by α + β - αβ and validate the approach on three QA datasets with five LLMs.

**Strengths:**

1. Unlike prior work (TRON, ConU), SAFER doesn't assume all instances can yield correct answers within finite sampling, incorporating a principled abstention mechanism that makes the framework more realistic for deployment.
2. The paper provides formal statistical guarantees with detailed proofs, properly applying the Clopper-Pearson exact method for finite-sample bounds and extending conformal risk control to the filtering stage.
3. The experimental validation is comprehensive
4. The filtering stage substantially reduces prediction set sizes (e.g., 7.9 to 5.5 for β=0.1 on TriviaQA) while maintaining statistical validity, improving practical usability.

**Weaknesses:**

1. The main components (Clopper-Pearson bounds, conformal risk control) are existing techniques. The contribution is primarily in their combination with abstention handling rather than fundamental methodological innovation.
2. Relies on exchangeability between calibration and test sets with no empirical evaluation of robustness to distribution shift
3. Requires users to specify both α (sampling risk) and β (filtering risk). The combined bound α + β - αβ is less interpretable than a single risk parameter
4. No principled guidance provided for parameter selection in practice
5. What fraction of instances require abstention? This is crucial for assessing practical utility
6. No wall-clock time comparisons or discussion of overhead
7. Only compares against TRON; missing comparisons with ConU, LofreeCP, SE-SCP, and other recent conformal prediction methods for open-ended QA.

**Questions:**

1.What are the abstention rates across different datasets, models, and risk levels? How does abstention correlate with model capability or question difficulty?
2. What is the wall-clock time cost compared to baseline sampling approaches? How does this scale with M and ŝ?
3. Can you provide practical guidance or heuristics for selecting α and β for different applications? What are reasonable default values?
4. How does SAFER perform when calibration and test distributions differ (e.g., temporal shift, domain shift)? This seems critical for real deployment.
5. How does SAFER compare quantitatively to ConU, LofreeCP, and other recent methods on the same datasets?
6. Have you evaluated alternative uncertainty measures? How much does performance depend on this choice?
7. How does the framework handle questions with multiple semantically distinct but correct answers?

---

> ### Author Response · Authors · 2025-11-21
> **Responses to Reviewer mzfb comments (Part 1)**
>
> > **W1:** The main components (Clopper-Pearson bounds, conformal risk control) are existing techniques. The contribution is primarily in their combination with abstention handling rather than fundamental methodological innovation.
>
> ***Response:*** We respectfully clarify that our innovation lies in identifying and resolving critical validity gaps in open-ended generation that naive applications overlook. While the mathematical components are established, their specific integration is motivated by two fundamental challenges that prior frameworks (e.g., ConU[2], TRON[3]) failed to address.
>
> First, we address the "Feasibility Gap" in finite sampling. Prior methods often overlook that specific risk levels may be inherently infeasible under fixed budgets. Motivated by this, we repurpose Clopper-Pearson to construct a novel feasibility test.
>
> Second, we address the "Filtering Gap" in noisy generation. As finite sampling yields initial sets mixed with incorrect answers, relying solely on uncalibrated heuristics cannot ensure valid risk control. We adapt CRC specifically as a calibrated filtering mechanism. This transforms CRC into a specialized selector for generation, strictly bridging the gap between imperfect scoring models and reliable coverage requirements.
>
>
>
> > **W2:** Relies on exchangeability between calibration and test sets with no empirical evaluation of robustness to distribution shift **Q4:** How does SAFER perform when calibration and test distributions differ (e.g., temporal shift, domain shift)? This seems critical for real deployment.
>
> ***Response:*** Existing methods that target coverage guarantees—such as LofreeCP [1], ConU, TRON, COPU [4], and SE-SCP [5]—are generally built upon the assumption of exchangeability, which is a foundational requirement of split conformal prediction [6].
>
>
>
> > **W3:** Requires users to specify both α (sampling risk) and β (filtering risk). The combined bound α + β - αβ is less interpretable than a single risk parameter
>
> ***Response:*** Given a fixed sampling budget, we can derive the minimum feasible risk level on the calibration set, based on which users can choose an appropriate value of $\alpha$. In conformal risk control frameworks, $\beta$ is typically set to 0.1 by default. The parameter α determines the test-time sampling size, whereas $\beta$ governs the size of the final prediction set.
>
>
>
> > **W4:** No principled guidance provided for parameter selection in practic
>
> ***Response:*** Sampling-based methods typically default to a fixed sampling size of 20. In our framework, however, we can determine the lower bound of the achievable risk level from the deployed calibration set, and then derive the test-time sampling size from the user-specified feasible risk level, i.e., $\alpha$. Because the risk level reflects user requirements, a smaller $\alpha$ necessitates a larger sampling budget. After the model is deployed, we can pre-compute the risk level, the corresponding test-time sampling size, and the uncertainty threshold on the local calibration set, enabling SAFER to perform inference efficiently at test time.
>
>
>
> > **W6:** No wall-clock time comparisons or discussion of overhead
>
> ***Response:*** SAFER can pre-compute the required sampling size and uncertainty threshold on the calibration set for any given $\alpha$ and $\beta$. At test time, once the user provides a prompt, these values can be applied directly.
>
> In terms of sampling cost, SAFER requires fewer generated answers than baselines such as TRON, which typically fix the sampling size to a constant (e.g., 20). Within a maximum sampling budget of 20, SAFER determines the minimum sampling size needed to satisfy the target risk level $\alpha$, achieving significantly lower overhead than existing methods.
>
>
> ---
>
> ### References
>
> [1] "Api is enough: Conformal prediction for large language models without logit-access." *arXiv preprint arXiv:2403.01216* (2024).
>
> [2] "Conu: Conformal uncertainty in large language models with correctness coverage guarantees." *Findings of the Association for Computational Linguistics: EMNLP 2024*. 2024.
>
> [3] "Sample then identify: A general framework for risk control and assessment in multimodal large language models." *arXiv preprint arXiv:2410.08174* (2024).
>
> [4] "Copu: Conformal prediction for uncertainty quantification in natural language generation." *arXiv preprint arXiv:2502.12601* (2025).
>
> [5]  "Addressing uncertainty in llms to enhance reliability in generative ai." *arXiv preprint arXiv:2411.02381* (2024).
>
> [6] "Conformal prediction: A gentle introduction." *Foundations and trends® in machine learning* 16.4 (2023): 494-591.

---

> ### Author Response · Authors · 2025-11-21
> **Responses to Reviewer mzfb comments (Part 2)**
>
> > **Q1:** What are the abstention rates across different datasets, models, and risk levels?
>
> ***Response:*** The abstention rate is influenced by model performance and task difficulty: stronger models and easier questions may yield a lower expected abstention rate, allowing for smaller feasible risk levels and thus enabling stricter coverage guarantees. We provide the abstention rates for open-book dataset CoQA (Table 1) and closed-book dataset TriviaQA (Table 2). As shown below, the abstention rate consistently decreases as the sampling budget ($M$) increases, and the empirical rate strictly remains below the theoretical upper bound across all settings, cosnfirming the framework's statistical validity.
>
> Table 1: Abstention rates and upper bounds on the CoQA dataset using Qwen2.5-14B and Qwen2.5-7B across varying sampling budgets ($M$).
>
> |  M   | Qwen2.5-14B (Rate$\pm$Std) | Upper Bound | Qwen2.5-7B (Rate$\pm$Std) | Upper Bound |
> | :--: | :------------------------: | :---------: | :-----------------------: | :---------: |
> |  1   |     0.3591$\pm$0.0113      |   0.3847    |     0.3722$\pm$0.0115     |   0.3980    |
> |  2   |     0.3076$\pm$0.0116      |   0.3320    |     0.3185$\pm$0.0115     |   0.3434    |
> |  3   |     0.2861$\pm$0.0112      |   0.3102    |     0.2960$\pm$0.0119     |   0.3201    |
> |  4   |     0.2711$\pm$0.0110      |   0.2946    |     0.2786$\pm$0.0116     |   0.3032    |
> |  5   |     0.2629$\pm$0.0108      |   0.2863    |     0.2695$\pm$0.0115     |   0.2936    |
> |  6   |     0.2584$\pm$0.0106      |   0.2821    |     0.2623$\pm$0.0115     |   0.2862    |
> |  7   |     0.2522$\pm$0.0107      |   0.2758    |     0.2558$\pm$0.0116     |   0.2798    |
> |  8   |     0.2477$\pm$0.0104      |   0.2706    |     0.2505$\pm$0.0116     |   0.2745    |
> |  9   |     0.2429$\pm$0.0103      |   0.2664    |     0.2467$\pm$0.0117     |   0.2703    |
> |  10  |     0.2398$\pm$0.0103      |   0.2633    |     0.2440$\pm$0.0117     |   0.2671    |
> |  11  |     0.2387$\pm$0.0104      |   0.2612    |     0.2404$\pm$0.0115     |   0.2639    |
> |  12  |     0.2341$\pm$0.0102      |   0.2570    |     0.2364$\pm$0.0114     |   0.2596    |
> |  13  |     0.2325$\pm$0.0101      |   0.2549    |     0.2349$\pm$0.0113     |   0.2575    |
> |  14  |     0.2306$\pm$0.0102      |   0.2528    |     0.2327$\pm$0.0111     |   0.2553    |
> |  15  |     0.2295$\pm$0.0103      |   0.2518    |     0.2300$\pm$0.0110     |   0.2532    |
> |  16  |     0.2271$\pm$0.0100      |   0.2497    |     0.2284$\pm$0.0111     |   0.2511    |
> |  17  |     0.2237$\pm$0.0099      |   0.2465    |     0.2242$\pm$0.0109     |   0.2468    |
> |  18  |     0.2219$\pm$0.0098      |   0.2444    |     0.2231$\pm$0.0109     |   0.2457    |
> |  19  |     0.2204$\pm$0.0096      |   0.2423    |     0.2231$\pm$0.0109     |   0.2457    |
>
> Table 2: Abstention rates and upper bounds on the TriviaQA dataset using Qwen2.5-14B model across varying sampling budgets ($M$).
>
> |  M   | Qwen2.5-14B (Rate$\pm$Std) | Upper Bound |
> | :--: | :------------------------: | :---------: |
> |  1   |     0.0509$\pm$0.0048      |   0.0633    |
> |  2   |     0.0289$\pm$0.0038      |   0.0394    |
> |  3   |     0.0211$\pm$0.0031      |   0.0295    |
> |  4   |     0.0165$\pm$0.0027      |   0.0244    |
> |  5   |     0.0133$\pm$0.0023      |   0.0206    |
> |  6   |     0.0123$\pm$0.0023      |   0.0193    |
> |  7   |     0.0117$\pm$0.0022      |   0.0193    |
> |  8   |     0.0117$\pm$0.0022      |   0.0193    |
> |  9   |     0.0112$\pm$0.0023      |   0.0180    |
> |  10  |     0.0112$\pm$0.0023      |   0.0180    |
> |  11  |     0.0097$\pm$0.0021      |   0.0166    |
> |  12  |     0.0086$\pm$0.0020      |   0.0153    |
> |  13  |     0.0086$\pm$0.0020      |   0.0153    |
> |  14  |     0.0080$\pm$0.0020      |   0.0139    |
> |  15  |     0.0074$\pm$0.0020      |   0.0125    |
> |  16  |     0.0074$\pm$0.0020      |   0.0125    |
> |  17  |     0.0074$\pm$0.0020      |   0.0125    |
> |  18  |     0.0074$\pm$0.0020      |   0.0125    |
> |  19  |     0.0074$\pm$0.0020      |   0.0125    |
>
> > **W5:** What fraction of instances require abstention? This is crucial for assessing practical utility
>
> ***Response:*** As shown in **Tables 1 and 2** above, the abstention fraction is strictly task-dependent. For tasks like **TriviaQA**, the rate is negligible (**< 1%**), indicating high practical utility. For tasks like **CoQA**, the rate is naturally higher but **decreases significantly** as the sampling budget increases. Crucially, these abstentions serve as a safety mechanism: they trigger only when the target risk is mathematically infeasible, thereby prioritizing validity over forced incorrect responses.

---

> ### Author Response · Authors · 2025-11-21
> **Responses to Reviewer mzfb comments (Part 3)**
>
> > **W7:** Only compares against TRON; missing comparisons with ConU, LofreeCP, SE-SCP, and other recent conformal prediction methods for open-ended QA. **Q5:** How does SAFER compare quantitatively to ConU, LofreeCP, and other recent methods on the same datasets?
>
> ***Response:*** ConU and SE-SCP both assume that the calibration and test samples can always obtain at least one correct answer within a finite sampling budget. This assumption is unrealistic, and under our setting these methods do not work. LofreeCP does not enforce a strict alignment between the nonconformity score and correctness during calibration, and thus cannot guarantee coverage at arbitrary user-specified risk levels during testing. We only compare against TRON because it explicitly considers whether the sampling set contains a correct answer; other related methods are not directly comparable. Our work is specifically designed to address the challenge of ensuring correct coverage under finite sampling while simultaneously filtering out incorrect answers from the sampling set without compromising the statistical guarantees.
>
>
>
> > **Q2:** What is the wall-clock time cost compared to baseline sampling approaches? How does this scale with M and ŝ?
>
> ***Response:*** Methods such as ConU typically fix the sampling size at 20. In contrast, SAFER can adjust the test-time sampling size based on the chosen $\alpha$. The maximum test-time sampling budget is 20, and as the allowed α increases, the required sampling size decreases.
>
>
>
> > **Q6:** Have you evaluated alternative uncertainty measures? How much does performance depend on this choice?
>
> ***Response:*** Our framework admits any uncertainty measure; the choice impacts only the **efficiency** (prediction set size), as the statistical **validity** is strictly guaranteed by conformal calibration regardless of the heuristic's quality. To demonstrate this flexibility with black-box models, we evaluated **GPT-4o-mini** on the **CoQA** dataset using a **consistency-based frequency score** as the uncertainty measure, confirming that SAFER maintains rigorous coverage even without access to internal model logits.
>
> Table 3: Empirical risk of Stage II on CoQA using GPT-4o-mini with consistency-based frequency scores, across varying risk levels ($\alpha$).
>
> | $\alpha$ |   Mean $\pm$ Std    |
> | :------: | :-----------------: |
> |   0.25   | 0.2293 $\pm$ 0.0173 |
> |   0.30   | 0.2462 $\pm$ 0.0229 |
> |   0.35   | 0.2745 $\pm$ 0.0234 |
> |   0.40   | 0.2934 $\pm$ 0.0110 |
> |   0.45   | 0.2960 $\pm$ 0.0036 |
> |   0.50   | 0.2960 $\pm$ 0.0036 |
> |   0.55   | 0.2960 $\pm$ 0.0036 |
>
> Table 4: Empirical risk of Stage II on CoQA using GPT-4o-mini with consistency-based frequency scores, with fixed $\alpha=0.25$ and varying total risk budgets.
>
> | $\alpha + \beta - \alpha\beta$ |   Mean $\pm$ Std    |
> | :----------------------------: | :-----------------: |
> |             0.2875             | 0.2293 $\pm$ 0.0173 |
> |             0.3250             | 0.2293 $\pm$ 0.0173 |
> |             0.3625             | 0.2293 $\pm$ 0.0173 |
> |             0.4000             | 0.2293 $\pm$ 0.0173 |
> |             0.4375             | 0.2674 $\pm$ 0.0834 |
> |             0.4750             | 0.3897 $\pm$ 0.1116 |
> |             0.5125             | 0.4691 $\pm$ 0.0852 |
> |             0.5500             | 0.5244 $\pm$ 0.0376 |
> |             0.5875             | 0.5570 $\pm$ 0.0323 |
> |             0.6250             | 0.5913 $\pm$ 0.0388 |
>
> > **Q3:** Can you provide practical guidance or heuristics for selecting α and β for different applications? What are reasonable default values?
>
> ***Response:*** Once the model and calibration set are prepared, we can compute the minimum feasible value of α under a given sampling budget. The user may then adjust $\alpha$ upward starting from this minimum. We can also pre-compute a mapping between different risk levels and their corresponding test-time sampling sizes, enabling the user to balance risk and cost. The parameter $\beta$ is obtained through the conformal risk control paradigm, and is typically set to a default value of 0.1[1].
>
>
>
> > **Q7:** How does the framework handle questions with multiple semantically distinct but correct answers?
>
> ***Response:*** Under the exchangeability assumption, the presence of linguistically non-equivalent yet correct samples in both the calibration and test sets does not violate the guarantee. Existing conformal prediction work has already addressed this type of multi-label scenario [2].
>
>
>
> ---
>
> ### References
>
> [1] Seeing and Reasoning with Confidence: Supercharging Multimodal LLMs with an Uncertainty-Aware Agentic Framework
>
> [2] Multi-label Classification under Uncertainty: A Tree-based Conformal Prediction Approach

---

> ### Comment · Reviewer_mzfb · 2025-11-28
>
> The responses provided by the authors are thorough and have improved my understandings of their methods. I don't have further clarification questions.

---

### Official Review · Reviewer_8PGK · 2025-10-31

**Soundness:** 3
**Presentation:** 2
**Contribution:** 2
**Rating:** 6
**Confidence:** 3

**Summary:**

This paper introduces SAFER, a two-stage framework for controlling the risk of miscoverage in open-ended question answering (QA) with Large Language Models (LLMs).  The core contribution is a practical and theoretically-grounded method for providing rigorous coverage guarantees for LLM outputs in open-ended QA, particularly by formally integrating an abstention option.

**Strengths:**

One of the main strengths of this work is its sound technical approach to a challenging problem. The formulation of a two-stage process that independently controls risk at both the sampling and filtering stages is elegant. The introduction of an explicit, statistically-grounded abstention mechanism is a significant step forward from prior methods that often assume an admissible answer is always attainable. This makes the framework more practical for real-world, risk-sensitive applications. The authors validate their method with experiments across multiple datasets, LLMs, and several different correctness evaluation metrics, demonstrating the robustness and data-efficiency of their framework.

**Weaknesses:**

A primary concern is the clarity of the novelty. While the application of abstention to this specific two-stage QA framework is new, the paper could better situate its contribution within the broader literature on conformal prediction and abstention, where similar concepts have been explored. The presentation is quite dense and assumes a high level of familiarity with conformal prediction, which may limit its accessibility to a wider audience. Furthermore, while the experiments are quite broad, they could be more rigorous. The analysis focuses on demonstrating that the empirical error rate is below the theoretical bounds, but a deeper analysis of the abstention cases or a more direct comparison with a wider set of non-conformal baselines would strengthen the claims of practical utility.

**Questions:**

Questions for the authors:

The abstention mechanism is a key component. Could you provide more insight into the practical trade-offs involved in setting the maximum sampling cap (M)? How sensitive is the framework's performance to this hyperparameter, and what is the relationship between M, the risk level ω, and the resulting calibrated sample size?
The filtering stage relies on an uncertainty measure U, which is sentence entropy in this work. How dependent is SAFER's performance on the quality of this uncertainty measure? Have you explored how the two-stage guarantees hold up when using other uncertainty heuristics, especially those that might be less correlated with correctness?

---

> ### Author Response · Authors · 2025-11-21
> **Responses to Reviewer 8PGK comments (Part 1)**
>
> > **W1:** A primary concern is the clarity of the novelty. While the application of abstention to this specific two-stage QA framework is new, the paper could better situate its contribution within the broader literature on conformal prediction and abstention, where similar concepts have been explored.
>
> ***Response:*** We respectfully clarify that our innovation lies in tailoring established mathematical components to resolve specific validity gaps in open-ended generation—nuances that prior works (e.g., ConU, TRON) overlooked. **Both stages of SAFER are driven by strong motivations to address these critical gaps**:
>
> Stage I addresses the "Feasibility Gap" via **(a) Risk Level Feasibility Verification and (b) Adaptive Test-Time Budgeting.** Prior works like ConU apply a fixed sampling budget, implicitly assuming a correct answer is always obtainable. Similarly, TRON fails to account for cases where the model cannot obtain an admissible answer on the calibration set even within the maximum sampling budget. This leads to invalid risk estimates for hard inputs. To address this, we use Clopper-Pearson intervals to (a) explicitly detect and abstain when the target risk is mathematically unattainable, avoiding the failure modes of prior methods. Furthermore, we (b) compute the *minimum* sampling budget required to meet the feasible risk. This adaptive strategy substantially reduces test-time costs while providing a principled way to utilize LTT .
>
> Stage II addresses the "Filtering Gap" via **(c) Statistically Rigorous Filtering.**. Since finite sampling yields initial sets mixed with incorrect answers, relying solely on heuristic uncertainty rankings often breaks statistical guarantees. To address this, we adapt Conformal Risk Control (CRC) to derive a statistically calibrated threshold. Unlike standard methods that discard answers based on arbitrary scores, our integration of CRC eliminates unreliable generations while strictly maintaining the user-specified coverage guarantee.
>
> > **W2:** The presentation is quite dense and assumes a high level of familiarity with conformal prediction, which may limit its accessibility to a wider audience.
>
> ***Response:*** We appreciate the feedback on accessibility. To support readers with varying backgrounds, we have explicitly included Appendix B ("Background of Split Conformal Prediction in Classification Tasks") as a self-contained primer. It details the foundational definitions and mechanisms of standard SCP.
>
> > **Q1:** The abstention mechanism is a key component. Could you provide more insight into the practical trade-offs involved in setting the maximum sampling cap (M)? **Q2:** How sensitive is the framework's performance to this hyperparameter, and what is the relationship between M, the risk level ω, and the resulting calibrated sample size?
>
> ***Response:*** The choice of $M$ balances **computational cost** against **abstention rate**. The abstention rate represents the proportion of samples failing to cover an admissible answer within the maximum budget M, effectively establishing the minimum feasible risk level for Stage I below which the system must abstain. As shown in the table below (Qwen2.5-14B on TriviaQA), the abstention rate drops from **5.1%** ($M=1$) to **0.74%** ($M=15$), where it fully saturates. Performance saturates at a dataset-specific threshold; increasing M beyond this point adds cost with zero utility gain.
>
> Table 1: Abstention Rates of Qwen2.5-14B on TriviaQA across sampling budgets ($M$).
> | $M$  |  Abstention Rate  | Upper Bound |
> | :--: | :---------------: | :---------: |
> |  1   | 0.0509$\pm$0.0048 |   0.0633    |
> |  2   | 0.0289$\pm$0.0038 |   0.0394    |
> |  3   | 0.0211$\pm$0.0031 |   0.0295    |
> |  4   | 0.0165$\pm$0.0027 |   0.0244    |
> |  5   | 0.0133$\pm$0.0023 |   0.0206    |
> |  6   | 0.0123$\pm$0.0023 |   0.0193    |
> |  7   | 0.0117$\pm$0.0022 |   0.0193    |
> |  8   | 0.0117$\pm$0.0022 |   0.0193    |
> |  9   | 0.0112$\pm$0.0023 |   0.0180    |
> |  10  | 0.0112$\pm$0.0023 |   0.0180    |
> |  11  | 0.0097$\pm$0.0021 |   0.0166    |
> |  12  | 0.0086$\pm$0.0020 |   0.0153    |
> |  13  | 0.0086$\pm$0.0020 |   0.0153    |
> |  14  | 0.0080$\pm$0.0020 |   0.0139    |
> |  15  | 0.0074$\pm$0.0020 |   0.0125    |
> |  16  | 0.0074$\pm$0.0020 |   0.0125    |
>
> > **W3:** The analysis focuses on demonstrating that the empirical error rate is below the theoretical bounds, but a deeper analysis of the abstention cases or a more direct comparison with a wider set of non-conformal baselines would strengthen the claims of practical utility.
>
> ***Response:*** We report detailed abstention rates in Table 1 above. Regarding practical utility, standard non-conformal baselines are analogous to the $M=1$ setting (single-sample generation). As shown in the Table 1 above (Qwen2.5-14B on TriviaQA), SAFER ($M=15$) significantly outperforms the single-sample baseline ($M=1$) by reducing the abstention rate from 5.1% ($M=1$) to 0.74% ($M=15$).

---

> ### Author Response · Authors · 2025-11-21
> **Responses to Reviewer 8PGK comments (Part 2)**
>
> > **Q3:** The filtering stage relies on an uncertainty measure U, which is sentence entropy in this work. How dependent is SAFER's performance on the quality of this uncertainty measure? **Q4:** Have you explored how the two-stage guarantees hold up when using other uncertainty heuristics, especially those that might be less correlated with correctness?
>
> Our framework admits any uncertainty measure. Crucially, the heuristic's quality affects **efficiency** (prediction set size) rather than **validity**, which remains statistically guaranteed. Stronger heuristics yield smaller, more informative sets, while weaker ones require larger sets to maintain coverage. We validated this flexibility with **GPT-4o-mini** on CoQA using **consistency-based frequency scores**, confirming rigorous coverage even in black-box settings without logit access.
>
> Table 2: Empirical risk of Stage II on CoQA using GPT-4o-mini with consistency-based frequency scores, across varying risk levels (α).
>
> Stage I:
>
> | $\alpha$ |   Mean $\pm$ Std    |
> | :------: | :-----------------: |
> |   0.25   | 0.2293 $\pm$ 0.0173 |
> |   0.30   | 0.2462 $\pm$ 0.0229 |
> |   0.35   | 0.2745 $\pm$ 0.0234 |
> |   0.40   | 0.2934 $\pm$ 0.0110 |
> |   0.45   | 0.2960 $\pm$ 0.0036 |
> |   0.50   | 0.2960 $\pm$ 0.0036 |
> |   0.55   | 0.2960 $\pm$ 0.0036 |
>
> Table 3: Empirical risk of Stage II on CoQA using GPT-4o-mini with consistency-based frequency scores, with fixed α=0.25 and varying total risk budgets.
>
> | $\alpha + \beta - \alpha\beta$ |   Mean $\pm$ Std    |
> | :----------------------------: | :-----------------: |
> |             0.2875             | 0.2293 $\pm$ 0.0173 |
> |             0.3250             | 0.2293 $\pm$ 0.0173 |
> |             0.3625             | 0.2293 $\pm$ 0.0173 |
> |             0.4000             | 0.2293 $\pm$ 0.0173 |
> |             0.4375             | 0.2674 $\pm$ 0.0834 |
> |             0.4750             | 0.3897 $\pm$ 0.1116 |
> |             0.5125             | 0.4691 $\pm$ 0.0852 |
> |             0.5500             | 0.5244 $\pm$ 0.0376 |
> |             0.5875             | 0.5570 $\pm$ 0.0323 |
> |             0.6250             | 0.5913 $\pm$ 0.0388 |

---

### Author Response · Authors · 2025-11-24
**Summary of Author Response to All the Reviewers**

We sincerely thank all reviewers for their insightful comments and constructive feedback. We have carefully revised our paper to address these concerns. In the revised PDF, **newly added or modified contents are marked in blue**, while **existing contents that may have been overlooked are marked in red**.

Our key updates and responses are summarized as follows:

**1. Additional Explaination**

- **Clarification of Contributions:** Our contributions focus on three components:  (a) determining whether a user-specified risk level is fundamentally feasible under finite sampling, (b) computing the minimum sampling budget required to meet the corresponding risk level, and (c) filtering unreliable samples by deriving a statistically calibrated uncertainty threshold.
- To enhance readability (addressing W2 from Reviewer 8PGK), we explicitly highlight **Appendix B**, which serves as a primer on Split Conformal Prediction to bridge the background gap for a wider audience.

**2. Additional Experimental Results**

- To demonstrate generalization beyond text-only QA (addressing W5 from Reviewer z2a5), we extended evaluations to the **MMVet** benchmark using **Qwen2-VL** and **InternVL2**. Results confirm that SAFER maintains strict risk control in complex multi-modal reasoning tasks (see **Appendix E.5**).
- To verify metric-agnosticism and API-only performance (addressing comments from reviewers z2a5 and mzfb), we evaluated **GPT-4o-mini** on CoQA using **consistency-based frequency scores** (without logit access). Results confirm SAFER’s validity even in black-box settings (see **Appendix E.3**).
- As requested by 8PGK, mzfb and z2a5, we provided a detailed analysis of the trade-off between the sampling cap M and abstention rates (see **Appendix E.4**).

**3. Additional Insights and Comparisons**

- Addressing the concerns of reviewer mzfb and z2a5, we restrict our comparison to TRON because other baselines (ConU, SE-SCP, LofreeCP) rely on unrealistic assumptions regarding finite sampling budgets or fail to theoretically guarantee coverage risks.
- As requested by reviewer z2a5, we also provided insights into the adaptive sampling budget mechanism and the theoretical trade-offs of joint calibration.

We are pleased to see the reviewers' acknowledgment of the contribution of our proposed method. We hope our explanations, additional experiments, and insights have adequately addressed your concerns. Please let us know if you have any further questions.

---

### Meta-Review · Area_Chair_fhm3 · 2026-01-06

**Summary:**

This paper introduces SAFER, a two-stage risk control framework for open-ended question answering with large language models that combines abstention-aware sampling and filtering. The method explicitly tests whether a target risk level is feasible under finite sampling using exact Clopper–Pearson bounds, abstaining when it is not, and adaptively determining the minimum sampling budget when it is. In a second stage, SAFER applies conformal risk control to filter unreliable generations while preserving statistical coverage guarantees. Experiments across multiple QA datasets, models, and evaluation criteria demonstrate that SAFER rigorously controls miscoverage risk and improves practical reliability compared to fixed-budget baselines. Authors have convinced reviewers about the novelty by clarifying more details and claims as well as adding new experiments.

**Reviewer Concerns:**

Reviewer 8PGK
The reviewer questioned the clarity of the paper’s novelty, noting that abstention and conformal methods exist in prior work and could be better contextualized. They also raised concerns about presentation density and requested deeper empirical analysis of abstention behavior and broader baseline comparisons.

Reviewer mzfb
This reviewer viewed the contribution as primarily an integration of existing statistical tools rather than a fundamentally new method, and highlighted missing discussion on robustness to distribution shift. Additional concerns included the interpretability of having two risk parameters, limited baseline comparisons, and the lack of wall-clock or computational cost analysis.

Reviewer z2a5
The reviewer considered the work incremental and questioned whether both stages amount to straightforward applications of known frameworks. They raised technical concerns about computational cost, the choice and exploration of the sampling cap (M), potential bias introduced by sequential calibration, and the absence of broader task domains and baseline methods in experiments.

**Reviewer Scores:**

Authors have convinced two reviewers to accept the paper. The last one does not participate in the discussion and has very low confidence in reviewing the paper. I believe they shared common questions and the last reviewer could have been convinced if active in discussion.

---

### Decision · Program_Chairs · 2026-01-26

Accept (Poster)